# A Large-Scale Atomic Interaction Model Based on Matter Wave Theory

## Abstract

Accurate and rapid prediction of atomic interactions constitutes a fundamental challenge in materials science. Traditional numerical methods face persistent limitations in balancing computational accuracy with efficiency. In contrast, AI-based large-scale atomic interaction models efficiently learn characteristic patterns of atomic configurations, enabling high-speed simulations while preserving accuracy. This offers a novel paradigm for molecular dynamics simulations and accelerated discovery of new materials and pharmaceuticals. To advance beyond current performance limits, this work proposes a matter wave theory-based large-scale atomic interaction model. First, we explicitly encode quantum mechanical matter wave theory into the neural network architecture, designing a quantum-inspired matter wave network as the core module. This innovation fundamentally enhances physical representation by effectively capturing atomic wave-particle duality. Subsequently, comprehensive error evaluation across multiple datasets (including Perovskite Oxides) demonstrates that our proposed Matter Wave Deep Potential Atomic model achieves root mean square errors of 0.5 meV/atom for energy and 28.7 meV/Å for force. These represent reductions of 16% and 8%, respectively, compared to state-of-the-art models including Deep Potential Atomic. Finally, as a standalone, general-purpose module, the matter wave network readily integrates with other advanced atomic interaction models. This adaptability will propel molecular dynamics simulation capabilities and expedite materials design and pharmaceutical discovery, thereby generating significant societal value.

## 1 Introduction

In materials science, predicting atomic interactions constitutes a fundamental research technology underpinning molecular dynamics simulations. This approach derives macroscopic material properties by simulating the evolution of atomic configurations in systems including small molecules, crystals, and clusters. Widely applied across computational physics, materials science, chemistry, and biology, it significantly accelerates development cycles while reducing costs for novel materials and pharmaceuticals. Early numerical methods for predicting atomic interactions faced inherent limitations in balancing computational accuracy with efficiency. These rigid, knowledge-driven paradigms incurred consistent computational costs even for similar configurations. For instance, Empirical Force Field (EFF) methods based on non-first-principles approaches rapidly compute interactions in large-scale atomic systems using physical properties and empirical parameters, yet suffer from insufficient accuracy. Conversely, first-principles Density Functional Theory (DFT) (Kohn & Sham, 1965) provides high-precision interaction data by solving the electronic Schrödinger equation from fundamental quantum mechanical principles. However, its computational inefficiency renders large-scale systems computationally prohibitive.

Recent advances in artificial intelligence have enabled a transformative data-driven computational paradigm. Machine learning models circumvent traditional limitations by concentrating computational costs primarily during training, where they learn mapping relationships between atomic configurations and interactions using finite samples and adjustable parameters. During inference, these models predict atomic interactions for similar configurations with minimal computational overhead. This approach effectively balances the accuracy-efficiency trade-off inherent in both EFF and DFT methods, enabling broad applications in materials design, drug discovery, and catalysis (Badaoui et al., 2022; Zeng et al., 2023a; Deringer et al., 2020; Wen et al., 2022; Ma & Liu, 2020; Yang

et al., 2022; Ng et al., 2023). Early machine learning potentials (Gastegger et al., 2018; Zhang et al., 2018a;b; Zeng et al., 2023b; Zhang et al., 2019) typically represent atomic configurations as graph structures, employing Message Passing Neural Network (MPNN) to capture interatomic correlations. While achieving near-DFT accuracy, these models face fundamental limitations: their training data require costly DFT computations, resulting in sparse datasets with limited diversity and inconsistent quality. Consequently, they exhibit poor generalization capabilities and significantly reduced predictive accuracy when encountering samples beyond their training distribution.

The accumulation of diverse datasets and standardization of data formats have facilitated the emergence of large-scale atomic interaction models (Schütt et al., 2021; Gasteiger et al., 2020; Takamoto et al., 2022; Zitnick et al., 2022; Shuaibi et al., 2021; Gasteiger et al., 2022; Liao et al., 2023; Batzner et al., 2022; Musaelian et al., 2023; Batatia et al., 2022; Zhang et al., 2024a;b). These models employ a transfer learning paradigm: pretraining on extensive datasets with varied atomic configurations using massively scalable parameters, followed by fine-tuning with limited task-specific samples. This approach enhances data-driven capabilities while effectively resolving generalization limitations. Furthermore, these architectures explicitly incorporate physical priors—including particle system equivariance, smoothness principles, and classical mechanics conservativeness—through structural encoding. This integration ensures adherence to fundamental physical laws and establishes a hybrid knowledge-data methodology. Consequently, such models demonstrate comprehensive improvements in predictive accuracy, parameter efficiency, and interpretability. However, the core architecture of leading models like Deep Potential Atomic (DPA2) (Zhang et al., 2024b) remains fundamentally based on standard Artificial Neural Network (ANN). While capable of capturing basic particle system behaviors and classical mechanics principles, these architectures struggle to represent intricate quantum mechanical phenomena, thereby limiting performance gains. Matter wave theory (De Broglie, 1924) provides essential prior knowledge for quantum-scale systems, establishing that all microscopic particles exhibit wave-particle duality through rigorous mathematical formalisms. Concurrently, the emerging Kolmogorov-Arnold Network (KAN) (Liu et al., 2024) architecture demonstrates superior capability in capturing complex quantum mechanical relationships compared to ANN, while achieving enhanced parameter efficiency and interpretability.

Building upon matter wave theory and the KAN architecture, this work enhances the state-of-the-art DPA2 model to propose the Matter Wave Deep Potential Atomic (WDPA) model. Our principal contributions are threefold:

1) We explicitly encode quantum mechanical matter wave theory into the neural network architecture, designing a quantum-inspired matter wave network as the core module. This innovation fundamentally advances theoretical representation by effectively capturing atomic wave-particle duality.

2) Comprehensive error evaluation across multiple datasets—including Perovskite Oxides (FerroEle) (Wu et al., 2023), Water Molecules (H2O-PD) (Zhang et al., 2021), Hafnium Dioxide (HfO2) (Zhang et al., 2024b), and Oxide Cathodes (Cathode) (Zhang et al., 2024b)—demonstrates that WDPA achieves root mean square errors of 0.5 meV/atom for energy and 28.7 meV/Å for force. These metrics represent reductions of 16% and 8%, respectively, compared to leading models including DPA2.

3) As a standalone, general-purpose module, the matter wave network enables seamless integration with other advanced atomic interaction models. This adaptability will accelerate developments in molecular dynamics simulation technologies and expedite materials design and pharmaceutical discovery pipelines, thereby generating significant scientific and societal impact.

## 2 RELATED WORK

The network architecture constitutes the core framework of large-scale atomic interaction models. Its capacity to capture feature distribution mappings from empirical systems decisively governs training convergence speed and fitting efficacy under constrained data and parameter conditions, thus directly determining predictive performance.

Explicit structural encoding of particle system equivariance as prior knowledge fundamentally enhances physical law adherence, establishing a hybrid knowledge-data paradigm that concurrently improves predictive accuracy, parameter efficiency, and interpretability. Equivariant Transformer (EquiformerV2) (Liao et al., 2023) exemplify this approach by processing atomic configurations through multiple equivariant constraints that preserve feature invariance under arbitrary translation, rotation, and permutation operations—thereby outperforming non-equivariant models like geomet-

ric Geometric Message Passing Neural Network (GemNet-OC) (Gasteiger et al., 2022). Large-scale interatomic potential models—including Neural Equivariant Interatomic Potentials (Nequip) (Batzner et al., 2022), Local Equivariant Interatomic Potentials (Allegro) (Musaelian et al., 2023), Message Passing Atomic Cluster Expansion (MACE) (Batatia et al., 2022), DPA1 (Zhang et al., 2024a), and DPA2 (Zhang et al., 2024b)—further demonstrate physical adherence through rigorous enforcement of particle system smoothness via continuous, exponentially decaying feature transitions and incorporation of classical mechanical conservativeness via force derivation from energy gradients rather than direct network fitting. This dual-physics encoding maintains molecular dynamics consistency while enhancing numerical stability. Specifically, DPA2 employs a sequential workflow: constructing atomic environments from coordinates and elemental types; encoding equivariance and smoothness constraints via its unique descriptor module; and fitting system energy before computing force, virial, and loss through energy gradients. This architecture reveals the unique descriptor as the performance-critical component for physical representation. However, its foundation in standard ANN with linear weight combinations and fixed activation functions fundamentally constrains quantum mechanical modeling—limiting performance gains despite successful classical physics encoding.

Within quantum mechanics (De Broglie, 1924), the uncertainty principle establishes the fundamental impossibility of simultaneous precise measurement of a microscopic particle's position and momentum, the existence of microscopic particles at any spacetime point can only be described probabilistically, rendering deterministic classical mechanics inadequate for quantum-scale phenomena. Matter wave theory resolves this limitation by demonstrating wave-like behavior in all microscopic particles, with wave-particle duality formally expressed through mathematical relationships to matter waves. Crucially, the wave function serves as the fundamental mathematical representation mapping classical mechanical properties—including energy, position, momentum, and time—to matter wave characteristics. Furthermore, the linear superposition principle governs multi-particle wave functions, while matter wave features determine the probabilistic existence of microscopic particles in spacetime and are dynamically correlated with their energy attributes, thereby forming a self-consistent convergence framework between matter wave functions and energy characteristics, establishing this framework as essential prior knowledge for quantum mechanical systems. Concurrently, the KAN (Liu et al., 2024) has emerged as a novel deep learning architecture derived from the Kolmogorov-Arnold representation theorem. For multivariate function approximation, KAN first decomposes target functions into univariate components according to the theorem, then fits these components using learnable activation functions before reconstructing outputs through linear summation and multilayer propagation. This design inherently captures matter wave features through properly designed activation functions, with linear summation and multilayer propagation analogous to the linear superposition principle and self-consistent convergence of matter wave functions, respectively. Notably, KAN achieves high predictive accuracy with significantly fewer parameters and enhanced interpretability, yielding superior parameter efficiency.

Thus, inspired by matter wave theory and the KAN architecture, we design a quantum-inspired matter wave network integrated into the unique descriptor of the DPA2 model, building upon MPNN. This integration enables strict adherence to quantum mechanical principles, significantly enhancing predictive performance. To fully capture atomic wave-particle duality per matter wave theory, the feature extraction process must be explicitly encoded within the neural network while approximating the linear superposition principle and self-consistent convergence of matter wave functions. Furthermore, potential information degradation during multi-level cascaded fusion necessitates optimized architectural connectivity.

To address these challenges, our methodology implements two key innovations: First, we design learnable activation functions derived from microscopic particle wave expressions. Through linear summation and multilayer propagation—which closely represent the linear superposition principle and self-consistent convergence of matter wave functions—these functions serve as MPNN aggregation operations to extract matter wave features and represent wave-particle duality. Second, we develop residual aggregation operations incorporating additive fusion mechanisms for MPNN combination operations. This design ensures complete informational integrity of matter wave features throughout hierarchical neural network fusion.

# 3 METHODOLOGY

First, the atomic environment is constructed based on the types and coordinate information of all atoms within the atomic configuration, yielding various atomic features. Next, an improved matter wave descriptor is employed to extract matter wave features, effectively capturing wave-particle duality, equivariance, and smoothness. Finally, the energy of the atomic configuration is fitted, and force, virial, and loss are computed using the energy gradient, thereby preserving conservativeness. The operational process of WDPA is illustrated in Figure 1.

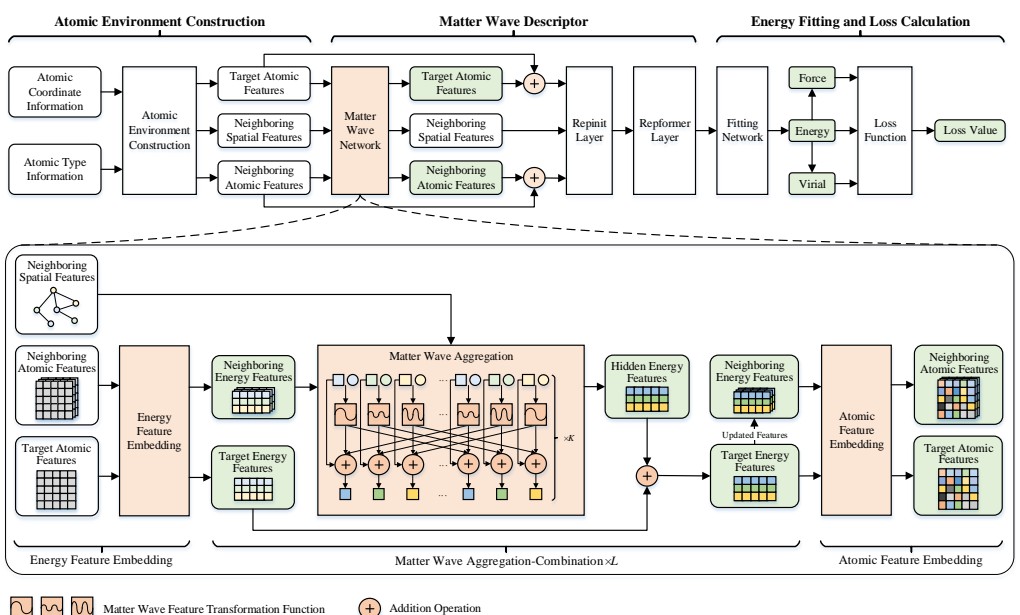

Figure 1: WDPA Computation Process Diagram. Upper: First, atomic features are constructed based on atomic coordinates and type information. These features are then updated using the matter wave network, followed by equivariant and smoothing operations through the repinit and repformer layers. Finally, the energy is fitted, and its negative gradient is computed to obtain the force, virial, and loss value. Bottom: The energy features are first computed using the energy feature embedding layer. Then, the hidden energy features are obtained through matter wave aggregation, and the energy features are updated through combination operations. Finally, atomic features are computed using the atomic feature embedding layer.

## 3.1 ATOMIC ENVIRONMENT CONSTRUCTION

The atomic configuration, consisting of $N$ atoms, is defined by its atomic type information $T \in \mathbb{R}^{N \times 1}$ and atomic coordinate information $C \in \mathbb{R}^{N \times 3}$. First, the atomic type information $T$ is converted into one-hot encoded features and processed through a MLP to generate the target atomic feature matrix $D \in \mathbb{R}^{n \times d}$, where $n$ represents the number of target atoms and $d$ denotes the number of features. Next, the atomic coordinate information $C$ is used to identify all neighboring atoms within the cutoff radius, constructing the neighboring atomic feature matrix $G \in \mathbb{R}^{n \times m \times d}$, where $m$ represents the number of neighboring atoms. Finally, to capture smoothness, the relative distances between each atom and its neighbors are calculated based on the atomic coordinate information $C$. These distances are processed using a switched inverse distance function (Zhang et al., 2024b), resulting in the neighboring spatial feature matrix $S \in \mathbb{R}^{n \times m \times 3}$.

## 3.2 MATTER WAVE DESCRIPTOR

### 3.2.1 MATTER WAVE NETWORK

First, energy feature embedding is performed for both the target atomic features and the neighboring atomic features. Next, based on the matter wave theory, matter wave aggregation-combination operations are applied in conjunction with neighboring spatial features. This process extracts the matter wave features of the atoms and captures the wave-particle duality of atoms. Finally, the target and neighboring atomic features are updated through atomic feature embedding.

**Energy Feature Embedding**   To represent atomic features as energy features in three spatial dimensions, a learnable energy feature embedding matrix $W^e \in \mathbb{R}^{d \times 3}$ is defined. The energy feature embedding operation is then applied to the target atomic feature matrix $D$ and the neighboring atomic feature matrix $G$, as expressed in Equation 1:

$$
\begin{aligned}
D^e &= D \cdot W^e, \\
G^e &= G \cdot W^e,
\end{aligned}
\tag{1}
$$

where $D^e \in \mathbb{R}^{n \times 3}$ represents the target energy feature matrix, while $G^e \in \mathbb{R}^{n \times m \times 3}$ denotes the neighboring energy feature matrix.

**Matter Wave Aggregation-Combination**   Consider a three-dimensional physical system centered on a target atom $i$ that includes multiple neighboring atoms $j$, as shown in Figure A1. Inspired by matter wave theory, according to the matter wave expression for microscopic particles, the energy feature value $G^e_j$ and spatial feature value $S_j$ of each neighboring atom $j$ determine its matter wave frequency $f = G^e_j / h$ and phase $\varphi = (p \cdot S_j)/h$ in each dimension, respectively. These parameters represent the wave-particle duality of each atom. Substituting the frequency and phase into the sine function to yield the matter wave feature transformation function $\Phi_j(\cdot)$ yields the expression shown in Equation 2:

$$
\Phi_j \left( G^e_j, S_j \right) = a \cdot \sin \left[ \frac{2\pi}{h} \cdot \left( G^e_j \cdot t + p \cdot S_j \right) \right],
\tag{2}
$$

where $a$, $t$, and $p$ represent unknown and adjustable amplitude, time, and momentum parameters, respectively, while $h$ denotes Planck's constant. Additionally, according to the linear superposition principle of matter wave functions, it follows that the summation of the matter wave feature transformation functions of all neighboring atoms can approximate their collective wave-particle duality. This summation is expressed as a total matter wave feature transformation function $\Phi^{total}(\cdot)$, which can take any arbitrarily complex form, as shown in Equation 3:

$$
\Phi^{total} \left( G^e, S \right) = \sum_{j=1}^{m} \Phi_j \left( G^e_j, S_j \right).
\tag{3}
$$

The total matter wave feature transformation function $\Phi^{total}(G^e, S)$ encapsulates the overall wave-particle duality of the physical system. Moreover, according to the self-consistent convergence of matter wave functions, the resulting matter wave feature exhibits an inherent correlation with the energy feature value $D^e_i$ of the target atom $i$ and $G^e_j$ of its neighboring atoms $j$.

First, to effectively transform the energy and spatial features of the target atom and its neighboring atoms using the matter wave feature transformation function, and to fully capture the wave-particle duality of atoms, an $L$-layer matter wave aggregation-combination operation is performed based on the MPNN structure. At this stage, the neighboring spatial feature matrix $S$ represents the relative distances between the target atom and its neighboring atoms across three spatial dimensions. Simultaneously, a $K$-layer matter wave aggregation operation is conducted, where the neighboring energy feature matrix $G^e$ is converted into the neighboring matter wave feature matrix $G^w \in \mathbb{R}^{n \times m \times 3}$. The matter wave feature transformation function $\Phi(\cdot)$, serving as a learnable activation function, is then applied to facilitate matter wave feature transformation, as expressed in Equation 4:

$$
\Phi_{l,k} \left( G^w_{l,k}, S \right) = A_{l,k} \odot \sin \left[ \frac{2\pi}{h} \cdot \left( G^w_{l,k} \odot T_{l,k} + P_{l,k} \odot S \right) \right],
\tag{4}
$$

where the matrix $A \in \mathbb{R}^{m \times 3}$ represents the learnable amplitude parameters, $T \in \mathbb{R}^{m \times 3}$ denotes the learnable time parameters, and $P \in \mathbb{R}^{m \times 3}$ corresponds to the learnable momentum parameters. Additionally, $l$ indicates the current layer of the matter wave aggregation-combination operation, while

$k$ represents the current layer of the matter wave aggregation operation, $\odot$ denotes the Hadamard product operation.

Next, based on the linear superposition principle and self-consistent convergence of matter wave functions, the neighboring matter wave feature transformation function results $\Phi_{l,k}(G_{l,k}^w, S)$ from the current layer are summed. Additionally, to preserve the integrity of matter wave features during multi-level cascaded fusion, residual connections are employed to transfer the neighboring matter wave feature information from the previous layer to the next layer's neighboring matter wave feature matrix $G_{l,k+1}^w \in \mathbb{R}^{n \times m \times 3}$, as expressed in Equation 5:

$$G_{l,k+1}^w = \tanh\left(W^1 \cdot \Phi_{l,k}\left(G_{l,k}^w, S\right) + G_{l,k}^w\right), \tag{5}$$

where the matrix $W^1 \in \mathbb{R}^{m \times m}$ is a constant matrix with all entries equal to 1, and $\tanh(\cdot)$ represents the activation function. Similarly, after multiple matter wave aggregation operations, the neighboring matter wave feature matrix at the $K$-th layer, $G_{l,K}^w \in \mathbb{R}^{n \times m \times 3}$, is obtained.

Finally, the neighboring matter wave feature matrix $G_{l,K}^w$ at the $K$-th layer is averaged along the dimension $m$, and the result is used as the hidden energy feature matrix $H_l^e \in \mathbb{R}^{n \times 3}$. Additionally, by using an additive operation, it is combined with the current layer's target energy feature matrix $D_l^e \in \mathbb{R}^{n \times 3}$ to produce the target energy feature matrix for the next layer, $D_{l+1}^e \in \mathbb{R}^{n \times 3}$, as shown in Equation 6:

$$
\begin{aligned}
H_l^e &= \tfrac{1}{m} \cdot \sum_{i=1}^{m} G_{l,K,i}^w, \\
D_{l+1}^e &= \tanh\left(H_l^e + D_l^e\right).
\end{aligned}
\tag{6}
$$

The neighboring energy feature matrix $G_{l+1}^e \in \mathbb{R}^{n \times m \times 3}$ is updated using the target energy feature matrix $D_{l+1}^e$. Similarly, after multiple matter wave aggregation-combination operations, the target energy feature matrix $D_L^e \in \mathbb{R}^{n \times 3}$ and the neighboring energy feature matrix $G_L^e \in \mathbb{R}^{n \times m \times 3}$ are obtained at the $L$-th layer.

**Atomic Feature Embedding** To update the energy features into atomic features, a learnable atomic feature embedding matrix $W^f \in \mathbb{R}^{3 \times d}$ is defined. The atomic feature embedding operation is then applied to the target energy feature matrix $D_L^e$ and the neighboring energy feature matrix $G_L^e$ at the $L$-th layer, as shown in Equation 7:

$$
\begin{aligned}
D^* &= D_L^e \cdot W^f, \\
G^* &= G_L^e \cdot W^f,
\end{aligned}
\tag{7}
$$

where $D^* \in \mathbb{R}^{n \times d}$ represents the updated target atomic feature matrix, and $G^* \in \mathbb{R}^{n \times m \times d}$ denotes the updated neighboring atomic feature matrix. The updated matrices will undergo residual connections with $D$ and $G$ respectively, and serve as the final output.

### 3.2.2 REPINIT LAYER AND REPFORMER LAYER

In the repinit layer (Zhang et al., 2024b), equivariant operations are performed using symmetrization operators based on the target atomic feature matrix $D^*$, neighboring atomic feature matrix $G^*$, and neighboring spatial feature matrix $S$. These operations explicitly represent the translational, rotational, and permutational invariance of atomic features. Additionally, to ensure smoothness, a switched inverse distance function is applied for smoothing, resulting in the updated target atomic feature matrix $D^p \in \mathbb{R}^{n \times d}$ and neighboring atomic feature matrix $G^p \in \mathbb{R}^{n \times m \times d}$.

In the repformer layer (Zhang et al., 2024b), multilayer equivariant and smoothing operations are performed based on the target atomic feature matrix $D^p$, neighboring atomic feature matrix $G^p$, and neighboring spatial feature matrix $S$, utilizing an attention mechanism. These operations deeply represent the equivariance and smoothness of atomic features. After feature filtering and fusion, the updated target atomic feature matrix $D^{pf} \in \mathbb{R}^{n \times d}$ is obtained.

### 3.3 ENERGY FITTING AND LOSS CALCULATION

First, a standard MLP and an energy bias layer are employed as the fitting network (Zhang et al., 2024b) to process the target atomic feature matrix $D^{pf}$ and extend it to multiple frames of the atomic configuration, resulting in the energy feature matrix $E \in \mathbb{R}^f$, where $f$ represents the number of frames. Next, to ensure conservativeness, the negative gradient of the energy feature matrix

$E$ is computed based on the atomic coordinate information $C$, yielding the force feature matrix $F \in \mathbb{R}^{f \times n \times 3}$ and the virial feature matrix $V \in \mathbb{R}^{f \times 9}$ for the atomic configuration. Finally, the loss value $L$ is computed based on the real energy, force, and virial feature matrices $E'$, $F'$, and $V'$ of the atomic configuration, as shown in Equation 8:

$$L = \frac{1}{f} \cdot \sum_{i=1}^{f} \left( \frac{p^e}{n} \cdot |E_i - E'_i|^2 + \frac{p^f}{3n} \cdot \sum_{j=1}^{n} \sum_{k=1}^{3} \left|F_{i,j,k} - F'_{i,j,k}\right|^2 + \frac{p^v}{9n} \cdot \sum_{j=1}^{9} \left|V_{i,j} - V'_{i,j}\right|^2 \right), \quad (8)$$

where the prefactor parameters for energy, force, and virial are denoted as $p^e$, $p^f$, and $p^v$, respectively.

## 4 EXPERIMENT

### 4.1 MODEL PARAMETERS

The evaluation experiments for WDPA, DPA2, and other models were conducted primarily on a cloud-based Nvidia A100 40GB GPU hardware environment. The maximum training time for a single run was approximately three days. In the WDPA model, the number of layers $L$ and $K$ were both set to 4, with training performed for either 1,000,000 steps or 3,000 steps. All other parameters were configured identically to those in DPA2 (Zhang et al., 2024b).

### 4.2 DATASET

Experiments were primarily conducted on publicly available datasets labeled with DFT calculations. The input data for all datasets comprised the types and coordinates of all atoms within the atomic configuration, while the output data included the energy, force, and virial of the atomic configuration. The datasets used are as follows:

1) **FerroEle** (Wu et al., 2023): This dataset contains DFT data of perovskite oxides, with atomic configuration comprising 15 types of atoms, including calcium, titanium, and oxygen. Each frame contains 20–60 atoms. The training set includes 6,966 frames, and the test set contains 760 frames.

2) **H2O-PD** (Zhang et al., 2021): A DFT dataset for water molecules, where the atomic configuration consists of hydrogen and oxygen atoms. Each frame contains 24–384 atoms. The training set comprises 46,077 frames, and the test set includes 2,342 frames.

3) **HfO2** (Zhang et al., 2024b): This dataset includes DFT data for hafnium dioxide, with atomic configuration consisting of oxygen and hafnium atoms. Each frame contains 96 atoms. The training set includes 27,660 frames, and the test set contains 917 frames.

4) **Cathode** (Zhang et al., 2024b): This dataset provides DFT data for oxide cathode materials used in lithium-ion and sodium-ion batteries. Atomic configuration is composed of 8 types of atoms, including lithium, sodium, and oxygen. Each frame contains 36–48 atoms. The training set includes 58,690 frames, and the test set contains 6,451 frames.

### 4.3 EVALUATION METRICS

The model's predictions for energy $E$ (total energy in meV, atomic energy in meV/atom), force $F$ (atomic force in meV/Å), and virial $V$ (total virial in meV, atomic virial in meV/atom) were evaluated using three primary metrics: Mean Absolute Error (MAE), Root Mean Square Error (RMSE), and the Coefficient of Determination ($R^2$). Let $M$ denote the number of entries in the flattened energy, force, or virial matrices, with $Y'$ representing the true values and $Y$ the predicted values. The specific evaluation methods are defined as follows:

1) **MAE**: The mean of the absolute errors between $Y'$ and $Y$, reflecting the accuracy of the predictions, as computed by Equation 9:

$$\text{MAE} = \frac{1}{M} \sum_{i=1}^{M} |Y'_i - Y_i|. \quad (9)$$

Table 1: Performance Comparison of Predictions (Datasets: FerroEle, H2O-PD, HfO2, Cathode).

| Metrics | Energy RMSE [meV/atom] | | | | Force RMSE [meV/Å] | | | |
| Models | FerroEle | H2O-PD | HfO2 | Cathode | FerroEle | H2O-PD | HfO2 | Cathode |
|---|---|---|---|---|---|---|---|---|
| **GNO** | 1.5 | OOM | 1.2 | 1.5 | 17.9 | OOM | 16.1 | 17.9 |
| **EFV-2** | 1.1 | OOM | 1.0 | 1.1 | **13.0** | OOM | **9.1** | 14.9 |
| **Nequip** | 1.1 | 0.9 | 1.5 | 14.3 | 23.0 | 27.1 | 58.8 | **14.3** |
| **Allegro** | 0.7 | OOM | 1.4 | 1.0 | 28.6 | OOM | 64.0 | 24.2 |
| **MACE** | 2.3 | 79.9 | 2.3 | 2.6 | 31.7 | 29.7 | 14.7 | 37.8 |
| **DPA2** | 0.6 | 0.6 | 0.9 | 1.0 | 31.4 | 28.1 | 54.7 | 29.7 |
| **WDPA** | **0.5** | **0.5** | **0.9** | **1.0** | 28.7 | **26.8** | 54.5 | 27.7 |

2) **RMSE**: The square root of the mean squared errors between $Y'$ and $Y$, reflecting the overall deviation of the predictions from the true values, as computed by Equation 10:

$$\text{RMSE} = \sqrt{\frac{1}{M} \sum_{i=1}^{M} (Y'_i - Y_i)^2}. \tag{10}$$

3) **$R^2$**: The proportion of the explained variance to the total variance of $Y'$ and $Y$, reflecting the overall fit of the model, as computed by Equation 11:

$$
\begin{aligned}
\text{TSS} &= \frac{1}{M} \sum_{i=1}^{M} \left(Y'_i - \overline{Y'}\right)^2, \\
\text{RSS} &= \frac{1}{M} \sum_{i=1}^{M} (Y'_i - Y_i)^2, \\
R^2 &= 1 - (\text{RSS}/\text{TSS}),
\end{aligned}
\tag{11}
$$

where RSS and TSS represent the unexplained variance and the total variance of the data, respectively.

## 4.4 COMPARATIVE EXPERIMENT

Pretraining experiments were conducted for 1 million steps using DPA2 and WDPA on the FerroEle, H2O-PD, HfO2, and Cathode datasets, and their performance was compared with other similar algorithms. The models involved in the experiments are as follows:

1) **GemNet-OC (GNO)** (Gasteiger et al., 2022): This model constructs a nearest neighbor graph and uses MPNN to capture the feature relationships between atoms and their neighboring atoms. It then uses two network output layers to predict energy and force.

2) **EquiformerV2 (EFV-2)** (Liao et al., 2023): This model is based on the Transformer architecture and employs equivariant operations to represent equivariance. It directly fits energy and force through a multi-layer network.

3) **Nequip** (Batzner et al., 2022): This model first performs equivariant operations and fits the energy of the atomic configuration using a network. The force is then obtained by calculating the negative gradient of the energy, thereby ensuring conservativeness.

4) **Allegro** (Musaelian et al., 2023): Allegro uses strict local equivariant operations to predict energy, offering superior prediction accuracy and scalability. It calculates force using the negative gradient of the energy.

5) **MACE** (Batatia et al., 2022): The model primarily employs an equivariant MPNN to predict energy, with atomic force derived as the negative gradient of the energy.

6) **DPA2** (Zhang et al., 2024b): This model uses attention mechanisms to perform multi-layer equivariant and smoothing operations, deepening the expression of equivariance and smoothness. It calculates force via the negative gradient of the energy.

7) **WDPA**: The model proposed in this paper.

The detailed prediction performance of the above models on all datasets is shown in Table 1, with "OOM" indicating memory overflow. The validation error curves during the training process for DPA2 and WDPA are shown in Figure A2, with the average errors summarized in Table 2.

As shown in Table 1, compared to GNO and EFV-2, WDPA exhibits smaller overall energy errors but larger force errors. This is primarily because WDPA adheres more closely to the principle of

Table 2: Comparison of Average Training and Validation Errors (Datasets: FerroEle, H2O-PD, HfO2, Cathode).

| Models / Metrics | | FerroEle DPA2 | FerroEle WDPA | H2O-PD DPA2 | H2O-PD WDPA | HfO2 DPA2 | HfO2 WDPA | Cathode DPA2 | Cathode WDPA |
|---|---|---|---|---|---|---|---|---|---|
| Energy RMSE | Train | 14.7 | **12.6** | 14.1 | **11.9** | 19.1 | **12.3** | 10.0 | **8.9** |
| [meV/atom] | Valid | 15.1 | **12.7** | 14.4 | **11.0** | 19.2 | **12.4** | 10.2 | **8.6** |
| Force RMSE | Train | 39.3 | **37.0** | 36.9 | **34.0** | 62.0 | **61.6** | 37.4 | **34.6** |
| [meV/Å] | Valid | 41.5 | **38.8** | 36.6 | **33.6** | 62.7 | **61.4** | 36.3 | **34.1** |

Table 3: Performance Comparison of Predictions (Datasets: FerroEle, H2O-PD, HfO2, Cathode).

| Models / Metrics | | FerroEle DPA2 | FerroEle WDPA | H2O-PD DPA2 | H2O-PD WDPA | HfO2 DPA2 | HfO2 WDPA | Cathode DPA2 | Cathode WDPA |
|---|---|---|---|---|---|---|---|---|---|
| Energy [meV] | MAE | 22.4 | **17.5** | 61.2 | **53.7** | 73.0 | **72.3** | 26.6 | **23.0** |
| | RMSE | 30.7 | **23.4** | 89.3 | **82.5** | 95.5 | **94.9** | 40.8 | **40.3** |
| | $R^2$(%) | 96.0 | **96.9** | 82.7 | **85.4** | 98.2 | 98.1 | **99.7** | 99.5 |
| Energy [meV/atom] | MAE | 0.5 | **0.4** | 0.4 | **0.4** | 0.7 | **0.7** | 0.6 | **0.5** |
| | RMSE | 0.6 | **0.5** | 0.6 | **0.5** | 0.9 | **0.9** | 1.0 | **1.0** |
| Force [meV/Å] | MAE | 22.2 | **20.2** | 18.6 | **17.7** | 39.0 | **38.9** | 18.2 | **16.7** |
| | RMSE | 31.4 | **28.7** | 28.1 | **26.8** | 54.7 | **54.5** | 29.7 | **27.7** |
| | $R^2$(%) | 99.8 | **99.8** | 99.9 | **99.9** | 99.9 | **99.9** | 99.6 | **99.6** |
| Virial [meV] | MAE | 240.8 | **229.9** | 584.4 | **565.8** | 1168.5 | **1157.7** | 504.2 | **502.3** |
| | RMSE | 359.8 | **344.9** | 1113.0 | **1075.2** | 1744.4 | **1728.7** | 804.9 | 837.1 |
| | $R^2$(%) | 99.6 | **99.7** | 98.2 | 98.0 | 99.9 | **99.9** | 98.1 | 97.9 |
| Virial [meV/atom] | MAE | 5.6 | **5.4** | 3.6 | **3.5** | 12.1 | **12.0** | 11.5 | 11.6 |
| | RMSE | 8.4 | **8.1** | 5.3 | **5.1** | 18.1 | **18.0** | 18.7 | 20.2 |

equivariance, effectively utilizing equivariant operations to represent the equivariance. This enables the model to better fit the underlying relationships between atomic types, coordinates, and energy. However, while directly fitting force, the energy-based negative gradient approach effectively expresses conservativeness, enhancing the stability of the model for molecular dynamics simulations. This, however, results in an indirect trade-off, where energy errors contribute to a decrease in force prediction accuracy.

Compared to Nequip, Allegro and MACE, WDPA exhibits smaller overall energy prediction errors, but slightly larger force prediction errors. This is primarily due to WDPA's use of an attention mechanism for multi-layer equivariant operations, which effectively captures deeper expressions of equivariance and improves energy prediction accuracy. However, compared to other models, the multi-layer smoothing operations in WDPA effectively express smoothness, further enhancing the model's stability for molecular dynamics simulations. Nonetheless, the complexity of the energy calculation process makes it challenging to obtain force predictions using the negative energy gradient method, leading to a slight loss in force prediction accuracy.

By analyzing Table 2 and Figure A2, it is evident that compared to DPA2, WDPA exhibits smaller overall error values as well as lower average training and validation errors during the training process. Additionally, the validation error curve for WDPA remains consistently lower. It is also observed that both models show varying degrees of overfitting after reaching a certain number of training steps, as indicated by a slight increase in the validation error curve toward the end of the training. Therefore, the best prediction results during the training process are selected for detailed comparison, with the specific predictive performance shown in Figure A3 and Table 3.

As demonstrated in Figure A3 and Table 3, WDPA achieves lower overall error metrics and higher $R^2$ values compared to DPA2. This improvement primarily stems from WDPA's architecture, which leverages matter wave theory through integrated MPNN networks to extract atomic matter wave features. By explicitly capturing quantum-mechanical wave-particle duality, our framework enhances predictive performance.

Table 4: Molecular Dynamics Simulation Results.

| Parameters | Diffusion Coefficient [Å²/ps] | | | | | ED [%] | | EF [%] | |
|---|---|---|---|---|---|---|---|---|---|
| | Ground Truth | PBE | PBE+vdW | DPA2 | WDPA | DPA2 | WDPA | DPA2 | WDPA |
| **330K, 50ps, D** | 0.186 | 0.015 | 0.041 | 0.078 | 0.081 | 0.238 | **0.075** | 0.083 | **0.071** |
| **330K, 5ps, D** | – | – | – | – | – | 0.182 | **0.066** | 0.049 | **0.047** |
| **275K, 5ps, H** | – | – | – | – | – | -0.204 | **-0.177** | 0.064 | **0.062** |
| **330K, 5ps, H** | – | – | – | – | – | **-0.077** | 0.116 | 0.057 | **0.054** |
| **500K, 5ps, H** | – | – | – | – | – | 0.743 | **0.693** | 0.207 | **0.206** |

## 4.5 MOLECULAR DYNAMICS SIMULATION

First, a simulation system containing 64 water molecules ($H_2O$) was constructed, and the WDPA and DPA2 models pretrained on the H2O-PD dataset were employed as the potential energy functions within the LAMMPS molecular dynamics software. Then, canonical ensemble simulations of 5 ps or 50 ps were performed with a time step of 0.0005 ps, using different initial temperatures ranging from 275 K to 500 K. To enhance numerical stability, following the simulation method (Zheng et al., 2018), hydrogen atoms (H) were partially replaced by deuterium atoms (D) in selected runs. Finally, the energy trajectories ($E^p$) obtained from the WDPA and DPA2 simulations were analyzed to evaluate Energy Drift (ED) and Energy Fluctuation (EF), as given in Equation 12:

$$\text{ED} = \frac{E^e - E^s}{|E^s|},$$
$$\text{EF} = \frac{\sigma(E^p)}{|\overline{E^p}|}, \tag{12}$$

where $\sigma(\cdot)$ denotes the standard deviation function, $E^s$ is the energy at the start of the simulation, and $E^e$ is the energy at the end of the simulation. Under similar simulation conditions, the Diffusion Coefficient was evaluated from atomic trajectories and compared with DFT (including PBE and PBE+vdW) calculation results and experimental measurements (Zheng et al., 2018). The molecular dynamics simulation results are summarized in Table 4. As shown in Table 4, compared to DPA2, ED and EF of WDPA are consistently closer to zero. This is primarily because WDPA can more accurately predict energy from atomic configurations, effectively mitigating the accumulation of energy errors and better adhering to energy conservation, thereby enhancing simulation stability. Moreover, the improved energy prediction further reduces force calculation errors, leading to more realistic atomic motion simulations. Consequently, the Diffusion Coefficient obtained with WDPA is closer to the experimentally measured values than those derived from PBE, PBE+vdW, or DPA2.

## 5 CONCLUSION

The WDPA model effectively predicts atomic interactions from configurations. Our approach designs learnable activation functions based on microscopic particle matter wave expressions, where linear summation and multilayer propagation closely represent the linear superposition principle and self-consistent convergence of matter wave functions. Concurrently, we implement residual aggregation operations with additive fusion mechanisms for combination operations, ensuring complete information preservation during multi-level cascaded fusion. By explicitly encoding quantum mechanical matter wave theory into the neural architecture, we establish a quantum-inspired matter wave network as the core module of large-scale atomic interaction models. This fundamental innovation achieves theoretical advancement through rigorous representation of wave-particle duality. Experimental validation confirms WDPA's enhanced predictive performance, demonstrating high-precision atomic interaction prediction from configuration data. Furthermore, the matter wave network operates as a standalone module compatible with other advanced atomic interaction models, enabling broad applications across materials science. Future work will encode the self-consistent Schrödinger equation solution process from DFT into the network architecture, further advancing WDPA's predictive performance.

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

# A  Appendix

## A.1  Methodology Details

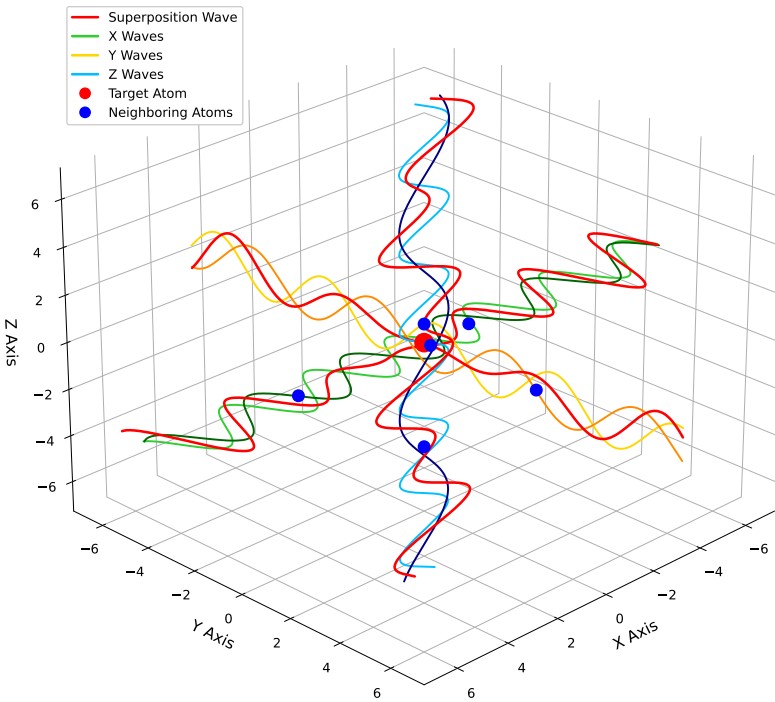

Figure A1: Demonstration of Three-Dimensional Matter Wave Feature Transformation Function Superposition.

Figure A1 illustrates the matter wave feature transformation function curves of individual atoms within a physical system, showcasing variations in amplitude, frequency, and phase across three dimensions. Additionally, it presents the resulting superposition of these wave functions.

## A.2  Comparative Experiment Details

The validation error curves of WDPA and DPA2 during the training process are compared in Figure A2. Subplots (a), (b), (c), and (d) illustrate the error results on the FerroEle, H2O-PD, HfO2, and Cathode datasets, respectively.

Figure A3 compares the prediction errors of WDPA and DPA2 across the FerroEle, H2O-PD, HfO2, and Cathode datasets.

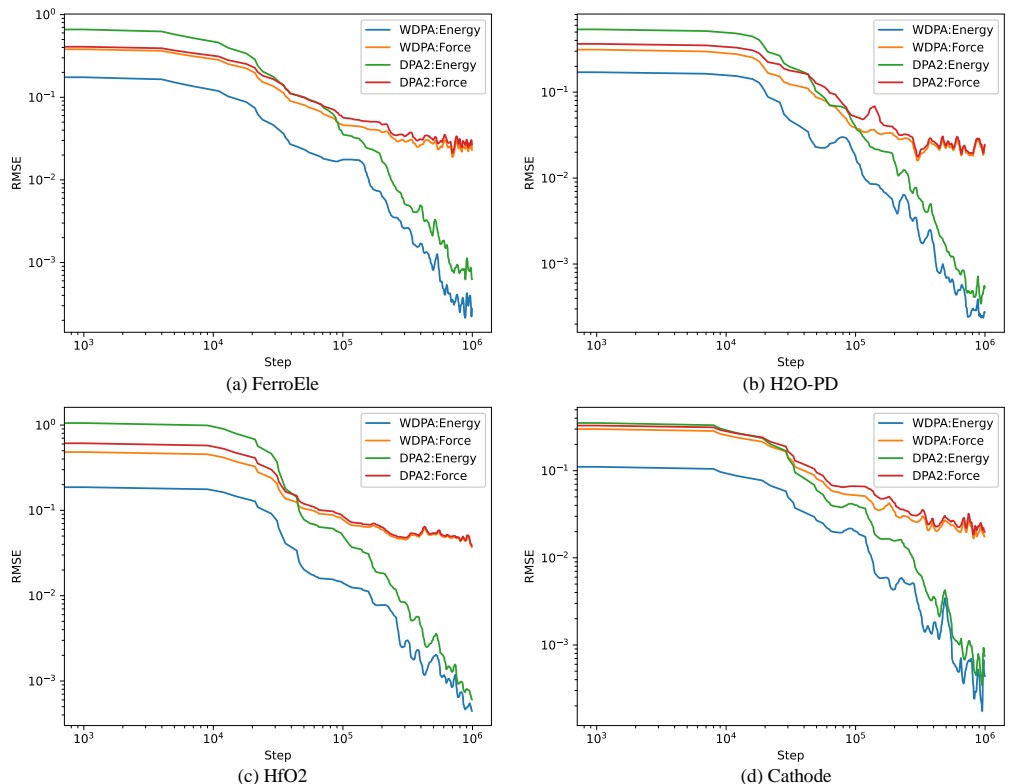

Figure A2: Validation Error Curve Comparison.

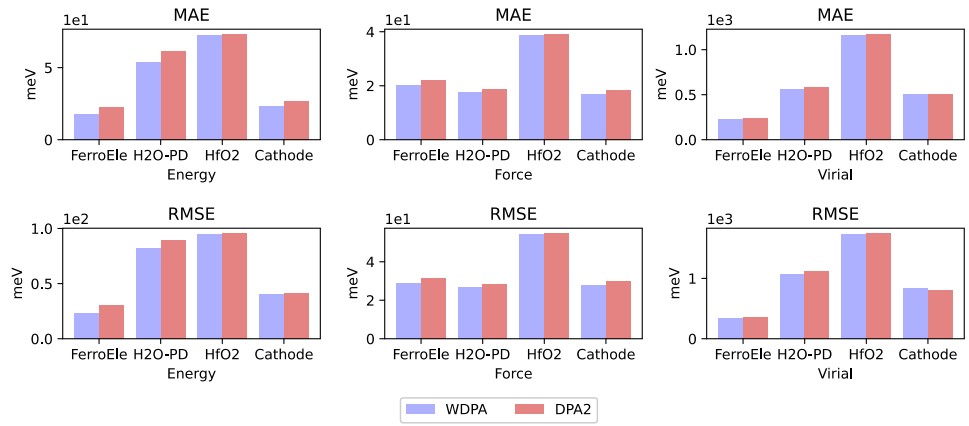

Figure A3: Comparative Prediction Error Analysis.

### A.3 ABLATION EXPERIMENT

#### A.3.1 DATASET

1) **H2O-PD-D**: A subset of the H2O-PD dataset, where each frame contains 192 atoms. The training set comprises 240 frames, and the test set includes 80 frames.

2) **FerroEle-D**: A subset of the FerroEle dataset, with each frame containing 60 atoms. The training set consists of 441 frames, and the test set includes 49 frames.

3) **SSE-PBE-D** (Huang et al., 2021): A DFT dataset for solid-state electrolytes. The atomic configuration is composed of lithium, germanium, phosphorus, sulfur, and silicon atoms. Each frame contains 400 atoms. The training set includes 391 frames, and the test set comprises 137 frames.

### A.3.2 RESULTS

We conducted an ablation study to evaluate the effects of the matter wave feature transformation function, residual connections, and combination operations within the WDPA architecture. The following experimental variants were examined:

1) **DPA2**: The baseline model.

2) **WDPA**: The model based on DPA2, enhanced with the matter wave network.

3) **WDPA-w**: Replaces the sinusoidal matter wave transformation function with elementary multiplication or addition operations.

4) **WDPA-p**: Several modifications to the matter wave feature transformation function were explored: replacing it with a Fourier function, decomposing its expression to separately transform energy and coordinate features, and removing Planck's constant.

5) **WDPA-r**: Modifies either the residual connections (through removal) or the combination operations (substituted with Hadamard product).

These models were pre-trained for 1 million steps or multiple 3000-step iterations on the FerroEle, H2O-PD-D, FerroEle-D, and SSE-PBE-D datasets. The best prediction results were selected for comparison, as shown in Figure A4 and Tables A1 and A2.

As shown in Figure A4 and Tables A1 and A2, compared to DPA2, WDPA-w, WDPA-p,

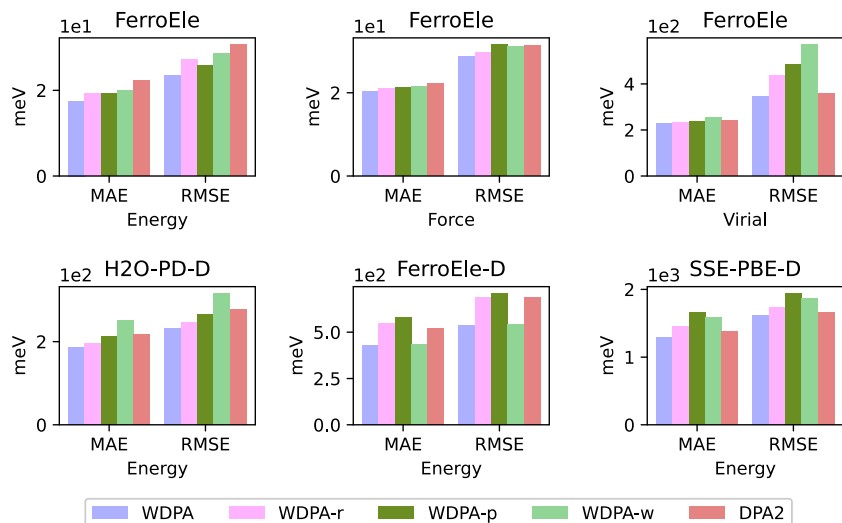

Figure A4: Comparative Prediction Error Analysis. This figure compares the prediction errors of WDPA, WDPA-r, WDPA-p, WDPA-w, and DPA2 across the FerroEle, H2O-PD-D, FerroEle-D, and SSE-PBE-D datasets.

and WDPA-r, WDPA exhibits smaller overall errors and higher $R^2$ values across multiple datasets, demonstrating a better ability to fit the underlying relationships between atomic types, coordinates, and energy, leading to more accurate predictions. This performance enhancement primarily arises from the model's ability to transform energy and spatial features of neighboring atoms into matter wave features using the matter wave feature transformation function, where linear summation and multilayer propagation closely represent the linear superposition principle and self-consistent convergence of matter wave functions, and in coordination with MPNN to fully express atomic wave-particle duality. Concurrently, residual aggregation operations with additive fusion mechanisms preserve feature integrity during multi-level cascaded fusion, collectively enhancing the model's predictive performance.

### A.4 ATOMIC CONFIGURATION OPTIMIZATION EXPERIMENT

First, stable atomic configurations of water molecules (H2O) containing 3 to 12 atoms were perturbed by applying Gaussian noise with standard deviations ranging from 0.1 to 0.5 std to generate initial atomic configurations. Subsequently, the model parameters of WDPA and DPA2, pretrained on the H2O-PD dataset, were frozen. The atomic coordinates in the input were treated as learnable

Table A1: Performance Comparison of Predictions (FerroEle Dataset, 1 Million Training Steps).

| Metrics \ Models | | DPA2 | WDPA-w | WDPA-p | WDPA-r | WDPA |
|---|---|---|---|---|---|---|
| Energy [meV] | MAE | 22.4 | 20.0 | 19.2 | 19.2 | **17.5** |
| | RMSE | 30.7 | 28.5 | 25.8 | 27.1 | **23.4** |
| | $R^2$(%) | 96.0 | 96.6 | 96.5 | 96.6 | **96.9** |
| Energy [meV/atom] | MAE | 0.5 | 0.4 | 0.4 | 0.4 | **0.4** |
| | RMSE | 0.6 | 0.6 | 0.5 | 0.5 | **0.5** |
| Force [meV/Å] | MAE | 22.2 | 21.6 | 21.3 | 20.9 | **20.2** |
| | RMSE | 31.4 | 31.0 | 31.6 | 29.6 | **28.7** |
| | $R^2$(%) | 99.8 | 99.8 | 99.8 | 99.8 | **99.8** |
| Virial [meV] | MAE | 240.8 | 253.9 | 237.6 | 232.7 | **229.9** |
| | RMSE | 359.8 | 571.1 | 485.0 | 434.8 | **344.9** |
| | $R^2$(%) | 99.6 | 99.5 | 99.5 | 99.6 | **99.7** |
| Virial [meV/atom] | MAE | 5.6 | 5.9 | 5.5 | 5.4 | **5.4** |
| | RMSE | 8.4 | 13.8 | 11.5 | 10.3 | **8.1** |

Table A2: Performance Comparison of Predictions (H2O-PD-D, FerroEle-D, and SSE-PBE-D Datasets, 3,000 Training Steps).

| Metrics \ Models | | | DPA2 | WDPA-w | WDPA-p | WDPA-r | WDPA |
|---|---|---|---|---|---|---|---|
| H2O-PD-D | Energy [meV] | MAE | 218.0 | 251.8 | 211.8 | 196.3 | **186.3** |
| | | RMSE | 277.8 | 316.4 | 266.6 | 246.0 | **232.4** |
| | | $R^2$(%) | 80.1 | 74.2 | 81.6 | 84.4 | **86.1** |
| | Energy [meV/atom] | MAE | 1.1 | 1.2 | 1.0 | 1.0 | **0.9** |
| | | RMSE | 1.4 | 1.6 | 1.3 | 1.2 | **1.2** |
| FerroEle-D | Energy [meV] | MAE | 522.1 | 434.5 | 581.0 | 548.8 | **429.1** |
| | | RMSE | 686.9 | 542.2 | 710.5 | 688.5 | **536.8** |
| | | $R^2$(%) | 88.3 | 92.7 | 85.3 | 88.2 | **92.8** |
| | Energy [meV/atom] | MAE | 8.7 | 7.2 | 9.6 | 9.1 | **7.1** |
| | | RMSE | 11.4 | 8.9 | 11.8 | 11.4 | **8.9** |
| SSE-PBE-D | Energy [meV] | MAE | 1382.8 | 1594.5 | 1664.2 | 1450.0 | **1295.6** |
| | | RMSE | 1654.3 | 1870.0 | 1944.1 | 1729.7 | **1614.9** |
| | | $R^2$(%) | 92.3 | 90.1 | 89.2 | 91.3 | **92.6** |
| | Energy [meV/atom] | MAE | 3.4 | 3.9 | 4.1 | 3.5 | **3.2** |
| | | RMSE | 4.1 | 4.6 | 4.8 | 4.2 | **4.0** |

parameters, and the energy output was used as the loss function for gradient descent optimization, thereby refining the coordinates until the minimum-energy stable atomic configuration was obtained. Finally, the energy (in eV) and average O–H bond lengths (in Å) of the optimized configurations were recorded. The experimental measurements of average O–H bond lengths (Curtiss et al., 2001) served as the reference for error analysis. For H2O configurations containing 3, 6, and 12 atoms, the numbers of optimization steps were set to 500, 2000, and 3000, respectively, with a learning rate of 0.01 applied throughout. The results of the atomic configuration optimization are summarized in Table A3. As shown in Table A3, compared to DPA2, the atomic configurations optimized by WDPA exhibit generally lower energy and error in average O–H bond length, indicating better convergence and results closer to the true stable atomic configurations. This improvement is attributed to the fact that the neural network design of WDPA incorporates prior knowledge from matter wave theory, making its computational process more consistent with physical laws. As a result, WDPA learns more realistic mapping relationships between atomic configurations and energy, demonstrating not only the potential for efficient atomic configuration optimization but also higher interpretability.

Table A3: Atomic Configuration Optimization Results.

| Atomic Configuration | Noise std | Average O-H Bond Length [Å] | | | Error [Å] | | Energy [eV] | |
|---|---|---|---|---|---|---|---|---|
| | | Ground Truth | DPA2 | WDPA | DPA2 | WDPA | DPA2 | WDPA |
| H2O (3 atoms) | 0.1 | 0.957 | 0.955 | 0.951 | **0.002** | 0.006 | -15.395 | **-15.639** |
| | 0.2 | 0.957 | 0.955 | 0.951 | **0.002** | 0.006 | -15.395 | **-15.640** |
| | 0.3 | 0.957 | 0.955 | 0.951 | **0.002** | 0.006 | -15.395 | **-15.639** |
| | 0.4 | 0.957 | 0.988 | 0.951 | 0.931 | **0.006** | -14.796 | **-15.640** |
| | 0.5 | 0.957 | >2.5 | 0.974 | >1.543 | **0.017** | **-14.637** | -14.548 |
| H2O (6 atoms) | 0.1 | ≈0.957 | 0.889 | 0.961 | ≈0.068 | ≈**0.004** | -29.408 | **-31.519** |
| | 0.2 | ≈0.957 | 0.889 | 0.962 | ≈0.068 | ≈**0.005** | -29.410 | **-31.360** |
| | 0.3 | ≈0.957 | 0.969 | 1.194 | ≈**0.012** | ≈0.237 | **-30.061** | -29.638 |
| | 0.4 | ≈0.957 | 0.976 | 0.972 | ≈0.019 | ≈**0.015** | -29.908 | **-30.207** |
| | 0.5 | ≈0.957 | 0.970 | 0.973 | ≈**0.013** | ≈0.016 | -29.985 | **-30.212** |
| H2O (12 atoms) | 0.1 | ≈0.957 | 1.066 | 1.062 | ≈0.109 | ≈**0.105** | -62.818 | **-62.986** |
| | 0.2 | ≈0.957 | 1.063 | 1.057 | ≈0.106 | ≈**0.100** | -62.824 | **-63.056** |
| | 0.3 | ≈0.957 | 1.056 | 1.060 | ≈**0.099** | ≈0.103 | -62.779 | **-63.063** |
| | 0.4 | ≈0.957 | 1.064 | 1.061 | ≈0.107 | ≈**0.104** | -62.815 | **-63.060** |
| | 0.5 | ≈0.957 | 1.064 | 1.062 | ≈0.107 | ≈**0.105** | -62.811 | **-63.059** |

