# OpenReview forum: "A Large-Scale Atomic Interaction Model Based on Matter Wave Theory"
_ICLR.cc/2026/Conference — Submitted to ICLR 2026_

### Official Review · Reviewer_KQgY · 2025-10-31

**Soundness:** 2
**Presentation:** 3
**Contribution:** 2
**Rating:** 2
**Confidence:** 4

**Summary:**

The paper proposes a new interatomic potential architecture inspired by matter-wave theory. It augments a deep potential network with sinusoidal descriptor blocks intended to encode quantum-wave-like periodicity in local atomic interactions. The model is trained on total energies with gradients for atomic forces and shows improved energy RMSE on four DFT datasets.

**Strengths:**

The sinusoidal block is a clear attempt to encode wave-based periodicity absent in standard local descriptors.

Interpretability potential: “Matter-wave” amplitudes could be correlated with bond-length or phase interference patterns.

Clear writing and figures: The pipeline and ablations are understandable, aiding reproducibility.

**Weaknesses:**

While promising, the evaluation remains narrow. To fully substantiate the physical motivation, the model should be tested on standard molecular (QM9, MD17/rMD17) and crystalline (Matbench, OC20/22, CrysMTM) benchmarks that probe diverse physics—small-molecule electronic properties, periodic-boundary effects, surfaces, and temperature- or phase-dependent behavior. Expanding targets beyond energy/forces to stress tensors, elastic moduli, phonon spectra, and electronic properties would reveal whether the “matter-wave” layer captures transferable physical features rather than dataset-specific correlations.

**Questions:**

Could the “matter-wave” descriptors be replaced by generic Fourier features—would accuracy degrade?

Does the model remain stable in long MD simulations (NVE drift, RDF/MSD consistency)?

---

> ### Author Response · Authors · 2025-11-20
>
> **Weakness 1:**
> >While promising, the evaluation remains narrow. To fully substantiate the physical motivation, the model should be tested on standard molecular (QM9, MD17/rMD17) and crystalline (Matbench, OC20/22, CrysMTM) benchmarks that probe diverse physics—small-molecule electronic properties, periodic-boundary effects, surfaces, and temperature- or phase-dependent behavior. Expanding targets beyond energy/forces to stress tensors, elastic moduli, phonon spectra, and electronic properties would reveal whether the “matter-wave” layer captures transferable physical features rather than dataset-specific correlations.
>
> **Response:**
> Thank you for providing valuable suggestions for improving the manuscript. The current research focuses on integrating prior knowledge from matter wave theory into the neural network design, validated on both fundamental Water Molecules datasets (H2O-PD) and datasets with potential application value, including Perovskite Oxides (FerroEle), Hafnium Dioxide (HfO2), and Oxide Cathodes (Cathode). These datasets are characterized by their relatively large system sizes, with individual atomic configurations ranging from tens to hundreds of atoms. Consequently, DFT labeling costs typically limit these datasets to several tens of thousands of frames. Compared to small-molecule datasets comprising hundreds of thousands to millions of frames—where individual configurations generally contain no more than 20 atoms—the present datasets offer advantages in terms of system complexity and specific application relevance. Future work will involve validation on additional datasets.
>
> **Question 1:**
> >Could the “matter-wave” descriptors be replaced by generic Fourier features—would accuracy degrade?
>
> **Response:**
> Thank you for providing valuable suggestions for improving the manuscript. To further validate the effectiveness of the matter wave feature transformation function, we replaced it with a Fourier function and decomposed the matter wave expression to separately transform energy and coordinate features or remove Planck's constant, denoted as the WDPA-p model, and added the model description in the original text in Section A.3.2, page 15, line 769-771 as follows:
>
> `4) WDPA-p: Several modifications to the matter wave feature transformation function were explored: replacing it with a Fourier function, decomposing its expression to separately transform energy and coordinate features, and removing Planck's constant.`
>
> We then conducted ablation studies on subsets of the Water Molecules, Perovskite Oxides, and Solid-State Electrolytes datasets—H2O-PD-D, FerroEle-D, and SSE-PBE-D—and supplemented the experimental results in the original text in Table A2 in Section A.3.2, page 16, line 840 as follows:
>
> ## Performance Comparison of Predictions
> **(H2O-PD-D, FerroEle-D, and SSE-PBE-D Datasets, 3,000 Training Steps)**
>
> | Dataset | Energy Type | Metric | DPA2 | WDPA-w | WDPA-p | WDPA-r | WDPA |
> |---------|-------------|--------|------|--------|------------|--------|----------|
> | H2O-PD-D | Energy [meV] | MAE | 218.0 | 251.8 | 211.8 | 196.3 | **186.3** |
> | | Energy [meV] | RMSE | 277.8 | 316.4 | 266.6 | 246.0 | **232.4** |
> | | Energy [meV] | R²(%) | 80.1 | 74.2 | 81.6 | 84.4 | **86.1** |
> | | Energy [meV/atom] | MAE | 1.1 | 1.2 | 1.0 | 1.0 | **0.9** |
> | | Energy [meV/atom] | RMSE | 1.4 | 1.6 | 1.3 | 1.2 | **1.2** |
> | FerroEle-D | Energy [meV] | MAE | 522.1 | 434.5 | 581.0 | 548.8 | **429.1** |
> | | Energy [meV] | RMSE | 686.9 | 542.2 | 710.5 | 688.5 | **536.8** |
> | | Energy [meV] | R²(%) | 88.3 | 92.7 | 85.3 | 88.2 | **92.8** |
> | | Energy [meV/atom] | MAE | 8.7 | 7.2 | 9.6 | 9.1 | **7.1** |
> | | Energy [meV/atom] | RMSE | 11.4 | 8.9 | 11.8 | 11.4 | **8.9** |
> | SSE-PBE-D | Energy [meV] | MAE | 1382.8 | 1594.5 | 1664.2 | 1450.0 | **1295.6** |
> | | Energy [meV] | RMSE | 1654.3 | 1870.0 | 1944.1 | 1729.7 | **1614.9** |
> | | Energy [meV] | R²(%) | 92.3 | 90.1 | 89.2 | 91.3 | **92.6** |
> | | Energy [meV/atom] | MAE | 3.4 | 3.9 | 4.1 | 3.5 | **3.2** |
> | | Energy [meV/atom] | RMSE | 4.1 | 4.6 | 4.8 | 4.2 | **4.0** |

---

> ### Author Response · Authors · 2025-11-20
>
> **Question 2:**
> >Does the model remain stable in long MD simulations (NVE drift, RDF/MSD consistency)?
>
> **Response:**
> Thank you for providing valuable suggestions for improving the manuscript. The core prerequisite for molecular dynamics (MD) simulations is achieving high-precision single-step energy and force predictions, while the stability effects of MD have been validated in the DPA2 literature [S.1]. This study primarily focuses on how integrating prior knowledge from matter wave theory improves the model's single-step prediction performance, and we will complete the MD experiments before the end of the discussion period.
>
> To further quickly validate the practical application effectiveness of the Matter Wave Deep Potential Atomic (WDPA) model, we have supplemented the original text with atomic configuration optimization experiments analogous to MD. Specifically, starting from noisy initial atomic configurations of water molecules (H2O), we used WDPA and DPA2 to iteratively optimize toward the minimum-energy stable atomic configuration, with error analysis conducted against experimentally measured O–H bond lengths (reference [S.2]) in Section 4.5, page 9, line 472-525 as follows:
>
> `4.5 Atomic Configuration Optimization Experiment`
>
> `First, stable atomic configurations of water molecules (H2O) containing 3 to 12 atoms were perturbed by applying Gaussian noise with standard deviations ranging from 0.1 to 0.5 std to generate initial atomic configurations... The results of the atomic configuration optimization are summarized in Table 4.`
>
> ## Atomic Configuration Optimization Results
>
> ### H2O (3 atoms)
>
> | Noise std | Length[Å] (GT) | Length[Å] (DPA2) | Length[Å] (WDPA) | Error[Å] (DPA2) | Error[Å] (WDPA) | Energy[eV] (DPA2) | Energy[eV] (WDPA) |
> |-----------|--------------|------|------|--------------|--------------|---------------|---------------|
> | 0.1 | 0.957 | 0.955 | 0.951 | **0.002** | 0.006 | -15.395 | **-15.639** |
> | 0.2 | 0.957 | 0.955 | 0.951 | **0.002** | 0.006 | -15.395 | **-15.640** |
> | 0.3 | 0.957 | 0.955 | 0.951 | **0.002** | 0.006 | -15.395 | **-15.639** |
> | 0.4 | 0.957 | 0.988 | 0.951 | 0.931 | **0.006** | -14.796 | **-15.640** |
> | 0.5 | 0.957 | >2.5 | 0.974 | >1.543 | **0.017** | **-14.637** | -14.548 |
>
> ### H2O (6 atoms)
>
> | Noise std | Length[Å] (GT) | Length[Å] (DPA2) | Length[Å] (WDPA) | Error[Å] (DPA2) | Error[Å] (WDPA) | Energy[eV] (DPA2) | Energy[eV] (WDPA) |
> |-----------|--------------|------|------|--------------|--------------|---------------|---------------|
> | 0.1 | ≈0.957 | 0.889 | 0.961 | ≈0.068 | **≈0.004** | -29.408 | **-31.519** |
> | 0.2 | ≈0.957 | 0.889 | 0.962 | ≈0.068 | **≈0.005** | -29.410 | **-31.360** |
> | 0.3 | ≈0.957 | 0.969 | 1.194 | **≈0.012** | ≈0.237 | **-30.061** | -29.638 |
> | 0.4 | ≈0.957 | 0.976 | 0.972 | ≈0.019 | **≈0.015** | -29.908 | **-30.207** |
> | 0.5 | ≈0.957 | 0.970 | 0.973 | **≈0.013** | ≈0.016 | -29.985 | **-30.212** |
>
> ### H2O (12 atoms)
>
> | Noise std | Length[Å] (GT) | Length[Å] (DPA2) | Length[Å] (WDPA) | Error[Å] (DPA2) | Error[Å] (WDPA) | Energy[eV] (DPA2) | Energy[eV] (WDPA) |
> |-----------|--------------|------|------|--------------|--------------|---------------|---------------|
> | 0.1 | ≈0.957 | 1.066 | 1.062 | ≈0.109 | **≈0.105** | -62.818 | **-62.986** |
> | 0.2 | ≈0.957 | 1.063 | 1.057 | ≈0.106 | **≈0.100** | -62.824 | **-63.056** |
> | 0.3 | ≈0.957 | 1.056 | 1.060 | **≈0.099** | ≈0.103 | -62.779 | **-63.063** |
> | 0.4 | ≈0.957 | 1.064 | 1.061 | ≈0.107 | **≈0.104** | -62.815 | **-63.060** |
> | 0.5 | ≈0.957 | 1.064 | 1.062 | ≈0.107 | **≈0.105** | -62.811 | **-63.059** |
>
> > **Note：**
> > - **GT** = Ground Truth
> > - **Length[Å]** = Average O-H Bond Length [Å]
>
> `As shown in Table 4, compared to DPA2, the atomic configurations optimized by WDPA exhibit generally lower energy and error in average O–H bond length, indicating better convergence and results closer to the true stable atomic configurations. This improvement is attributed to the fact that the neural network design of WDPA incorporates prior knowledge from matter wave theory, making its computational process more consistent with physical laws. As a result, WDPA learns more realistic mapping relationships between atomic configurations and energy, demonstrating not only the potential for efficient atomic configuration optimization but also higher interpretability.`
>
> [S.1] Zhang, D., Liu, X., Zhang, X., Zhang, C., Cai, C., Bi, H., Du, Y., Qin, X., Peng, A., Huang, J., others, 2024. DPA-2: a large atomic model as a multi-task learner. npj Computational Materials 10, 293. https://doi.org/10.1038/s41524-024-01493-2
>
> [S.2] Curtiss, L.A., Redfern, P.C., Raghavachari, K., Pople, J.A., 2001. Gaussian-3X (G3X) theory: Use of improved geometries, zero-point energies, and Hartree–Fock basis sets. The Journal of Chemical Physics 114, 108–117.

---

> ### Author Response · Authors · 2025-12-03
> **Response to Reviewer KQgY (Second Round)**
>
> **Question 1:**
> >Could the “matter-wave” descriptors be replaced by generic Fourier features—would accuracy degrade?
>
> **Response:**
> Based on our initial response, to further validate the effectiveness of the matter wave function, ablation studies were conducted on the perovskite oxide dataset FerroEle, and the experimental results were added to the original text in Table A1 in Section A.3.2, page 16, line 810 as follows:
>
> ## Performance Comparison of Predictions
> **(FerroEle Dataset, 1 Million Training Steps)**
>
> | Metrics / Models | DPA2 | WDPA-w | WDPA-p | WDPA-r | **WDPA** |
> | :--- | :---: | :---: | :---: | :---: | :---: |
> | **Energy [meV]** | | | | | |
> | &nbsp;&nbsp;MAE | 22.4 | 20.0 | 19.2 | 19.2 | **17.5** |
> | &nbsp;&nbsp;RMSE | 30.7 | 28.5 | 25.8 | 27.1 | **23.4** |
> | &nbsp;&nbsp;R²(%) | 96.0 | 96.6 | 96.5 | 96.6 | **96.9** |
> | **Energy [meV/atom]** | | | | | |
> | &nbsp;&nbsp;MAE | 0.5 | 0.4 | 0.4 | 0.4 | **0.4** |
> | &nbsp;&nbsp;RMSE | 0.6 | 0.6 | 0.5 | 0.5 | **0.5** |
> | **Force [meV/Å]** | | | | | |
> | &nbsp;&nbsp;MAE | 22.2 | 21.6 | 21.3 | 20.9 | **20.2** |
> | &nbsp;&nbsp;RMSE | 31.4 | 31.0 | 31.6 | 29.6 | **28.7** |
> | &nbsp;&nbsp;R²(%) | 99.8 | 99.8 | 99.8 | 99.8 | **99.8** |
> | **Virial [meV]** | | | | | |
> | &nbsp;&nbsp;MAE | 240.8 | 253.9 | 237.6 | 232.7 | **229.9** |
> | &nbsp;&nbsp;RMSE | 359.8 | 571.1 | 485.0 | 434.8 | **344.9** |
> | &nbsp;&nbsp;R²(%) | 99.6 | 99.5 | 99.5 | 99.6 | **99.7** |
> | **Virial [meV/atom]** | | | | | |
> | &nbsp;&nbsp;MAE | 5.6 | 5.9 | 5.5 | 5.4 | **5.4** |
> | &nbsp;&nbsp;RMSE | 8.4 | 13.8 | 11.5 | 10.3 | **8.1** |
>
> Concurrently, the visualization of the table was updated in Figure A4 in Section A.3.2, page 15, line 774.
>
> **Question 2:**
> >Does the model remain stable in long MD simulations (NVE drift, RDF/MSD consistency)?
>
> **Response:**
> Based on our initial response, molecular dynamics (MD) simulations have been added to the original text. Specifically, using a water molecule (H2O) simulation system, WDPA and DPA2 were employed as the potential energy functions in the LAMMPS molecular dynamics software. The simulations were used to evaluate energy drift, energy fluctuation, and diffusion coefficient, with the latter being compared against both experimental measurements and DFT calculation results (reference [S.1]) in Section 4.5, page 10, line 497-518 as follows:
>
> `4.5 Molecular Dynamics Simulation`
>
> `First, a simulation system containing 64 water molecules (H2O) was constructed, and the WDPA and DPA2 models pretrained on the H2O-PD dataset were employed as the potential energy functions within the LAMMPS molecular dynamics software... The molecular dynamics simulation results are summarized in Table 4.`
>
> ## Molecular Dynamics Simulation Results.
>
> ### Diffusion Coefficient [Å²/ps]
>
> | Parameters         | Ground Truth | PBE   | PBE+vdW | DPA2  | WDPA  |
> | :----------------- | :----------: | :---: | :-----: | :---: | :---: |
> | **330K, 50ps, D**  |    0.186     | 0.015 |  0.041  | 0.078 | 0.081 |
>
> ### Energy Drift (ED) [%]
>
> | Parameters         | DPA2   | WDPA      |
> | :----------------- | :----: | :-------: |
> | **330K, 50ps, D**  | 0.238  | **0.075** |
> | **330K, 5ps, D**   | 0.182  | **0.066** |
> | **275K, 5ps, H**   | -0.204 | **-0.177**|
> | **330K, 5ps, H**   | **-0.077** | 0.116  |
> | **500K, 5ps, H**   | 0.743  | **0.693** |
>
> ### Energy Fluctuation (EF) [%]
>
> | Parameters         | DPA2  | WDPA      |
> | :----------------- | :---: | :-------: |
> | **330K, 50ps, D**  | 0.083 | **0.071** |
> | **330K, 5ps, D**   | 0.049 | **0.047** |
> | **275K, 5ps, H**   | 0.064 | **0.062** |
> | **330K, 5ps, H**   | 0.057 | **0.054** |
> | **500K, 5ps, H**   | 0.207 | **0.206** |
>
> `As shown in Table 4, compared to DPA2, ED and EF of WDPA are consistently closer to zero... Consequently, the diffusion coefficient obtained with WDPA is closer to the experimentally measured values than those derived from PBE, PBE+vdW, or DPA2.`
>
> The content related to the atomic configuration optimization experiments has been moved to the appendix in Section A.4, pages 15-17, lines 806-900.
>
> [S.1] Zheng, L., Chen, M., Sun, Z., Ko, H.-Y., Santra, B., Dhuvad, P., Wu, X., 2018. Structural, electronic, and dynamical properties of liquid water by ab initio molecular dynamics based on SCAN functional within the canonical ensemble. The Journal of Chemical Physics 148.

---

### Official Review · Reviewer_AD9s · 2025-11-01

**Soundness:** 1
**Presentation:** 3
**Contribution:** 2
**Rating:** 2
**Confidence:** 4

**Summary:**

This paper proposed a new machine learning force field model inspired by the matter wave theory and Kolmogorov-Arnold Network (KAN). The core of the method is a feature map $\Phi(G, S) = A \odot \sin [\frac{2\pi}{h} \cdot (P\cdot S + G\cdot T)]$ inspired by the de Broglie wave $\psi(\mathbf{r},t)=A e^{i(\mathbf{k}\cdot \mathbf{r} - \omega t)}$. Here $G$ is a energy feature and $S$ is a spatial feature, and momentum $P$ and time $T$ are learnable weights. The new network is modified from the previous work DPA2 by adding this matter wave feature, and energy and force prediction experiments was conducted on 4 different dataset against baselnie methods.

**Strengths:**

The paper is well-written and easy to follow. Figure 1 explains the overall architecture pretty well. The experiment seems comprehensive, and the model performs well in energy prediction on the chosen datasets.

**Weaknesses:**

- The performance improvements are marginal at best. For force prediction, it is actually worse than other methods, except for the H2O-PD dataset.
- matter wave theory is a *linear theory*. Since you have multiple layers with nonlinear activation, I would say calling this a matter wave is a stretch. This is more like a Fourier feature.
- The paper asserts that linear aggregation "satisfies" superposition and "self‑consistent convergence" (Sec. 2–3.2) but gives no derivation or quantitative test that the learned features actually behave like wavefunctions (e.g., interference patterns under controlled setups, dispersion relations)
- The usage of KAN is not well motivated. There are no ablation studies demonstrating that using KAN is beneficial.
- Using sinusoids to encode geometry is not new. atomistic GNNs often use periodic/basis expansions (e.g., Bessel/spherical‑harmonic bases), and many models use Fourier features or sinusoidal activations. The paper does not benchmark against such periodic encodings, which weakens a novelty claim tied to "periodicity."
- Units inconsistency. Table 2 labels "Energy RMSE [meV/atom]" yet reports values in the 10–20 meV/atom range during training/validation, while Table 1 shows 0.5–1.0 meV/atom test RMSE for the same datasets. This doesn't make any sense.
- Several baselines are marked "OOM" without describing the memory setting, batch sizes, or whether mixed precision and gradient checkpointing were used—this can bias comparisons.
- Hyperparameters critical for reproducibility are missing.
- There is no molecular‑dynamics evaluation (e.g., NVE energy drift, long‑horizon stability, diffusion coefficients). Since the motivation is improved MD, the per‑frame test error is only a proxy.

**Questions:**

- Why don't you learn $W^{1}$ in equation 5? Since you are already learning momentum and time, setting this weighting to uniformly $1$ is strange.
- Why did you cite Kohn-Sham instead of de Broglie for the matter wave theory? This is very strange to me.
- From the experiments, WDPA for energy prediction works much better than force prediction. Do you have any explanation for this?
- Why do you need Planck's constant here? There is no inherent physical process here, so we can set the scale to whatever we like. In fact, in most DFT calculations we use atomic units where $h=1$.

---

> ### Author Response · Authors · 2025-11-20
>
> **Weakness 1:**
> >The performance improvements are marginal at best. For force prediction, it is actually worse than other methods, except for the H2O-PD dataset.
>
> **Response:**
> Thank you for providing valuable suggestions for improving the manuscript. Table 1 in the original text shows that the force prediction error of the Matter Wave Deep Potential Atomic (WDPA) model is significantly reduced compared to the Deep Potential Atomic (DPA2) model. The reason why the force prediction performance of the WDPA model does not reach the optimum is that it calculates forces via the negative gradient of energy, which introduces error accumulation. Consequently, its single-step force prediction performance is inferior to models like the Equivariant Transformer (EquiformerV2) that employ direct force fitting. However, the advantage of computing forces through the energy gradient lies in its inherent adherence to physical laws, which enhances the stability of long-time molecular dynamics simulations, as validated in the DPA2 model literature [S.1]. We have elaborated on the relevant content in the original text in Section 4.4, page 8, line 427-468.
>
> [S.1] Zhang, D., Liu, X., Zhang, X., Zhang, C., Cai, C., Bi, H., Du, Y., Qin, X., Peng, A., Huang, J., others, 2024. DPA-2: a large atomic model as a multi-task learner. npj Computational Materials 10, 293. https://doi.org/10.1038/s41524-024-01493-2
>
> **Weakness 2:**
> >matter wave theory is a linear theory. Since you have multiple layers with nonlinear activation, I would say calling this a matter wave is a stretch. This is more like a Fourier feature.
>
> **Response:**
> Thank you for providing valuable suggestions for improving the manuscript. Our design of the matter wave network is primarily inspired by matter wave theory, rather than strictly adhering to its physical principles. Therefore, when approximating the linear superposition principle through linear summation of matter wave features, we introduced nonlinear activation functions for engineering considerations to enhance practical prediction performance. For the question regarding similarities with Fourier features, please refer to our response to Weakness 5.
>
> **Weakness 3:**
> >The paper asserts that linear aggregation "satisfies" superposition and "self consistent convergence" (Sec. 2–3.2) but gives no derivation or quantitative test that the learned features actually behave like wavefunctions (e.g., interference patterns under controlled setups, dispersion relations)
>
> **Response:**
> Thank you for providing valuable suggestions for improving the manuscript. We acknowledge that it is indeed inappropriate to equate the linear superposition principle and self-consistent convergence characteristics directly with the linear summation and multilayer propagation operations in the matter wave network. Accordingly, we have revised the original wording from "satisfy" to " analogous to" or " closely represent" in relevant sections. Nevertheless, this approximate relationship has genuinely inspired the design of the matter wave network, and we have made a preliminary attempt to establish this correspondence to lay a foundation for future research on interpretability.
>
> **Weakness 4:**
> >The usage of KAN is not well motivated. There are no ablation studies demonstrating that using KAN is beneficial.
>
> **Response:**
> Thank you for providing valuable suggestions for improving the manuscript. Our approach is mainly inspired by the Kolmogorov-Arnold Network (KAN) model, where we redesign the learnable activation functions as matter wave feature transformation functions to approximate the matter wave feature transformation in matter wave theory. We have elaborated on the relevant content in the original text in Section 2, page 3, line 144-158.

---

> ### Author Response · Authors · 2025-11-20
>
> **Weakness 5:**
> >Using sinusoids to encode geometry is not new. atomistic GNNs often use periodic/basis expansions (e.g., Bessel/spherical harmonic bases), and many models use Fourier features or sinusoidal activations. The paper does not benchmark against such periodic encodings, which weakens a novelty claim tied to "periodicity."
>
> **Response:**
> Thank you for providing valuable suggestions for improving the manuscript. Fourier and Bessel radial-angular basis functions, along with positional encoding, serve as effective representation methods, and the matter wave feature transformation function shares certain similarities with them. However, a key distinction lies in the fact that the matter wave feature transformation function explicitly captures the relationship between energy features and spatial features, rather than merely representing spatial features alone. Its design is inspired by the core relational expressions in matter wave theory—namely, the matter wave frequency $f = \frac{E}{h}$ and wavelength $\lambda  = \frac{h}{p}$, where $E$, $p$, and $h$ denote energy, momentum, and Planck’s constant, respectively. By substituting these into a concise sinusoidal wave function, we guide the model to learn underlying physical laws. To preserve the completeness of this formulation, Planck’s constant is retained, as reflected in the matter wave feature transformation function described in the original text. Theoretically, these expressions can be incorporated into any form of wave function, since the core insight lies in the unique relational expressions for matter wave frequency and wavelength, rather than in the specific form of the trial wave function. For instance, one may also adopt non-periodic forms such as wavelet functions, rather than being limited to periodic representations like Fourier basis functions.
>
> To further validate the effectiveness of the matter wave feature transformation function, we replaced it with a Fourier function and decomposed the matter wave expression to separately transform energy and coordinate features or remove Planck's constant, denoted as the WDPA-p model, and added the model description in the original text in Section A.3.2, page 15, line 769-771 as follows:
>
> `4) WDPA-p: Several modifications to the matter wave feature transformation function were explored: replacing it with a Fourier function, decomposing its expression to separately transform energy and coordinate features, and removing Planck's constant.`
>
> We then conducted ablation studies on subsets of the Water Molecules, Perovskite Oxides, and Solid-State Electrolytes datasets—H2O-PD-D, FerroEle-D, and SSE-PBE-D—and supplemented the experimental results in the original text in Table A2 in Section A.3.2, page 16, line 840 as follows:
>
> ## Performance Comparison of Predictions
> **(H2O-PD-D, FerroEle-D, and SSE-PBE-D Datasets, 3,000 Training Steps)**
>
> | Dataset | Energy Type | Metric | DPA2 | WDPA-w | WDPA-p | WDPA-r | WDPA |
> |---------|-------------|--------|------|--------|------------|--------|----------|
> | H2O-PD-D | Energy [meV] | MAE | 218.0 | 251.8 | 211.8 | 196.3 | **186.3** |
> | | Energy [meV] | RMSE | 277.8 | 316.4 | 266.6 | 246.0 | **232.4** |
> | | Energy [meV] | R²(%) | 80.1 | 74.2 | 81.6 | 84.4 | **86.1** |
> | | Energy [meV/atom] | MAE | 1.1 | 1.2 | 1.0 | 1.0 | **0.9** |
> | | Energy [meV/atom] | RMSE | 1.4 | 1.6 | 1.3 | 1.2 | **1.2** |
> | FerroEle-D | Energy [meV] | MAE | 522.1 | 434.5 | 581.0 | 548.8 | **429.1** |
> | | Energy [meV] | RMSE | 686.9 | 542.2 | 710.5 | 688.5 | **536.8** |
> | | Energy [meV] | R²(%) | 88.3 | 92.7 | 85.3 | 88.2 | **92.8** |
> | | Energy [meV/atom] | MAE | 8.7 | 7.2 | 9.6 | 9.1 | **7.1** |
> | | Energy [meV/atom] | RMSE | 11.4 | 8.9 | 11.8 | 11.4 | **8.9** |
> | SSE-PBE-D | Energy [meV] | MAE | 1382.8 | 1594.5 | 1664.2 | 1450.0 | **1295.6** |
> | | Energy [meV] | RMSE | 1654.3 | 1870.0 | 1944.1 | 1729.7 | **1614.9** |
> | | Energy [meV] | R²(%) | 92.3 | 90.1 | 89.2 | 91.3 | **92.6** |
> | | Energy [meV/atom] | MAE | 3.4 | 3.9 | 4.1 | 3.5 | **3.2** |
> | | Energy [meV/atom] | RMSE | 4.1 | 4.6 | 4.8 | 4.2 | **4.0** |
>
> **Weakness 6:**
> >Units inconsistency. Table 2 labels "Energy RMSE [meV/atom]" yet reports values in the 10–20 meV/atom range during training/validation, while Table 1 shows 0.5–1.0 meV/atom test RMSE for the same datasets. This doesn't make any sense.
>
> **Response:**
> Thank you for providing valuable suggestions for improving the manuscript. Table 2 in the original text presents the average prediction errors over the entire training process, while Table 1 reports the final prediction errors after model convergence. Therefore, the error ranges in Table 1 are naturally lower than those in Table 2, and the units of measurement are consistent across both tables.

---

> ### Author Response · Authors · 2025-11-20
>
> **Weakness 7:**
> >Several baselines are marked "OOM" without describing the memory setting, batch sizes, or whether mixed precision and gradient checkpointing were used—this can bias comparisons.
>
> **Response:**
> Thank you for providing valuable suggestions for improving the manuscript. The "OOM" data are cited from the DPA2 model literature [S.1], and the experimental configurations of other comparative models, including memory usage and batch size, can be referred to in the DPA2 model literature [S.1].
>
> To enable a more comprehensive comparison of model performance, we have added an introduction to the Message Passing Atomic Cluster Expansion (MACE) model [S.2] and cited it in Section 4.4, page 8, line 411 as follows:
>
> `5) MACE (Batatia et al., 2022): The model primarily employs an equivariant MPNN to predict energy, with atomic force derived as the negative gradient of the energy.`
>
> We have also supplemented the prediction error data of the MACE model in Table 1 in Section 4.4, page 9, line 440 as follows:
>
> ## Performance Comparison of Predictions
> **(Datasets: FerroEle, H2O-PD, HfO2, Cathode)**
>
> ### Energy RMSE [meV/atom]
>
> | Models | FerroEle | H2O-PD | HfO2 | Cathode |
> |--------|----------|--------|------|---------|
> | GNO | 1.5 | OOM | 1.2 | 1.5 |
> | EFV-2 | 1.1 | OOM | 1.0 | 1.1 |
> | Nequip | 1.1 | 0.9 | 1.5 | 14.3 |
> | Allegro | 0.7 | OOM | 1.4 | 1.0 |
> | MACE | 2.3 | 79.9 | 2.3 | 2.6 |
> | DPA2 | 0.6 | 0.6 | 0.9 | 1.0 |
> | WDPA | **0.5** | **0.5** | **0.9** | **1.0** |
>
> ### Force RMSE [meV/Å]
>
> | Models | FerroEle | H2O-PD | HfO2 | Cathode |
> |--------|----------|--------|------|---------|
> | GNO | 17.9 | OOM | 16.1 | 17.9 |
> | EFV-2 | **13.0** | OOM | **9.1** | 14.9 |
> | Nequip | 23.0 | 27.1 | 58.8 | **14.3** |
> | Allegro | 28.6 | OOM | 64.0 | 24.2 |
> | MACE | 31.7 | 29.7 | 14.7 | 37.8 |
> | DPA2 | 31.4 | 28.1 | 54.7 | 29.7 |
> | WDPA | 28.7 | **26.8** | 54.5 | 27.7 |
>
> [S.1] Zhang, D., Liu, X., Zhang, X., Zhang, C., Cai, C., Bi, H., Du, Y., Qin, X., Peng, A., Huang, J., others, 2024. DPA-2: a large atomic model as a multi-task learner. npj Computational Materials 10, 293. https://doi.org/10.1038/s41524-024-01493-2
>
> [S.2] Batatia, I., Kovacs, D.P., Simm, G., Ortner, C., Csányi, G., 2022. MACE: Higher order equivariant message passing neural networks for fast and accurate force fields. Advances in neural information processing systems 35, 11423–11436.
>
> **Weakness 8:**
> >Hyperparameters critical for reproducibility are missing.
>
> **Response:**
> Thank you for providing valuable suggestions for improving the manuscript. Apart from the hyperparameters of the matter wave network, all other hyperparameters remain consistent with those in the DPA2 model literature [S.1]. We have stated and cited this in the original text in Section 4.1, page 7, line 341-345.
>
> [S.1] Zhang, D., Liu, X., Zhang, X., Zhang, C., Cai, C., Bi, H., Du, Y., Qin, X., Peng, A., Huang, J., others, 2024. DPA-2: a large atomic model as a multi-task learner. npj Computational Materials 10, 293. https://doi.org/10.1038/s41524-024-01493-2

---

> ### Author Response · Authors · 2025-11-20
>
> **Weakness 9:**
> >There is no molecular dynamics evaluation (e.g., NVE energy drift, long horizon stability, diffusion coefficients). Since the motivation is improved MD, the per frame test error is only a proxy.
>
> **Response:**
> Thank you for providing valuable suggestions for improving the manuscript. The core prerequisite for molecular dynamics (MD) simulations is achieving high-precision single-step energy and force predictions, while the stability effects of MD have been validated in the DPA2 literature [S.1]. This study primarily focuses on how integrating prior knowledge from matter wave theory improves the model's single-step prediction performance, and we will complete the MD experiments before the end of the discussion period.
>
> To further quickly validate the practical application effectiveness of the Matter Wave Deep Potential Atomic (WDPA) model, we have supplemented the original text with atomic configuration optimization experiments analogous to MD. Specifically, starting from noisy initial atomic configurations of water molecules (H2O), we used WDPA and DPA2 to iteratively optimize toward the minimum-energy stable atomic configuration, with error analysis conducted against experimentally measured O–H bond lengths (reference [S.3]) in Section 4.5, page 9, line 472-525 as follows:
>
> `4.5 Atomic Configuration Optimization Experiment`
>
> `First, stable atomic configurations of water molecules (H2O) containing 3 to 12 atoms were perturbed by applying Gaussian noise with standard deviations ranging from 0.1 to 0.5 std to generate initial atomic configurations... The results of the atomic configuration optimization are summarized in Table 4.`
>
> ## Atomic Configuration Optimization Results
>
> ### H2O (3 atoms)
>
> | Noise std | Length[Å] (GT) | Length[Å] (DPA2) | Length[Å] (WDPA) | Error[Å] (DPA2) | Error[Å] (WDPA) | Energy[eV] (DPA2) | Energy[eV] (WDPA) |
> |-----------|--------------|------|------|--------------|--------------|---------------|---------------|
> | 0.1 | 0.957 | 0.955 | 0.951 | **0.002** | 0.006 | -15.395 | **-15.639** |
> | 0.2 | 0.957 | 0.955 | 0.951 | **0.002** | 0.006 | -15.395 | **-15.640** |
> | 0.3 | 0.957 | 0.955 | 0.951 | **0.002** | 0.006 | -15.395 | **-15.639** |
> | 0.4 | 0.957 | 0.988 | 0.951 | 0.931 | **0.006** | -14.796 | **-15.640** |
> | 0.5 | 0.957 | >2.5 | 0.974 | >1.543 | **0.017** | **-14.637** | -14.548 |
>
> ### H2O (6 atoms)
>
> | Noise std | Length[Å] (GT) | Length[Å] (DPA2) | Length[Å] (WDPA) | Error[Å] (DPA2) | Error[Å] (WDPA) | Energy[eV] (DPA2) | Energy[eV] (WDPA) |
> |-----------|--------------|------|------|--------------|--------------|---------------|---------------|
> | 0.1 | ≈0.957 | 0.889 | 0.961 | ≈0.068 | **≈0.004** | -29.408 | **-31.519** |
> | 0.2 | ≈0.957 | 0.889 | 0.962 | ≈0.068 | **≈0.005** | -29.410 | **-31.360** |
> | 0.3 | ≈0.957 | 0.969 | 1.194 | **≈0.012** | ≈0.237 | **-30.061** | -29.638 |
> | 0.4 | ≈0.957 | 0.976 | 0.972 | ≈0.019 | **≈0.015** | -29.908 | **-30.207** |
> | 0.5 | ≈0.957 | 0.970 | 0.973 | **≈0.013** | ≈0.016 | -29.985 | **-30.212** |
>
> ### H2O (12 atoms)
>
> | Noise std | Length[Å] (GT) | Length[Å] (DPA2) | Length[Å] (WDPA) | Error[Å] (DPA2) | Error[Å] (WDPA) | Energy[eV] (DPA2) | Energy[eV] (WDPA) |
> |-----------|--------------|------|------|--------------|--------------|---------------|---------------|
> | 0.1 | ≈0.957 | 1.066 | 1.062 | ≈0.109 | **≈0.105** | -62.818 | **-62.986** |
> | 0.2 | ≈0.957 | 1.063 | 1.057 | ≈0.106 | **≈0.100** | -62.824 | **-63.056** |
> | 0.3 | ≈0.957 | 1.056 | 1.060 | **≈0.099** | ≈0.103 | -62.779 | **-63.063** |
> | 0.4 | ≈0.957 | 1.064 | 1.061 | ≈0.107 | **≈0.104** | -62.815 | **-63.060** |
> | 0.5 | ≈0.957 | 1.064 | 1.062 | ≈0.107 | **≈0.105** | -62.811 | **-63.059** |
>
> > **Note：**
> > - **GT** = Ground Truth
> > - **Length[Å]** = Average O-H Bond Length [Å]
>
> `As shown in Table 4, compared to DPA2, the atomic configurations optimized by WDPA exhibit generally lower energy and error in average O–H bond length, indicating better convergence and results closer to the true stable atomic configurations. This improvement is attributed to the fact that the neural network design of WDPA incorporates prior knowledge from matter wave theory, making its computational process more consistent with physical laws. As a result, WDPA learns more realistic mapping relationships between atomic configurations and energy, demonstrating not only the potential for efficient atomic configuration optimization but also higher interpretability.`
>
> [S.1] Zhang, D., Liu, X., Zhang, X., Zhang, C., Cai, C., Bi, H., Du, Y., Qin, X., Peng, A., Huang, J., others, 2024. DPA-2: a large atomic model as a multi-task learner. npj Computational Materials 10, 293. https://doi.org/10.1038/s41524-024-01493-2
>
> [S.3] Curtiss, L.A., Redfern, P.C., Raghavachari, K., Pople, J.A., 2001. Gaussian-3X (G3X) theory: Use of improved geometries, zero-point energies, and Hartree–Fock basis sets. The Journal of Chemical Physics 114, 108–117.

---

> ### Author Response · Authors · 2025-11-20
>
> **Question 1:**
> >Why don't you learn ${W^1}$ in equation 5? Since you are already learning momentum and time, setting this weighting to uniformly 1 is strange.
>
> **Response:**
> Thank you for providing valuable suggestions for improving the manuscript. The matrix ${W^1}$ is introduced to perform linear summation through multiplication with the matter wave feature matrix.
>
> **Question 2:**
> >Why did you cite Kohn-Sham instead of de Broglie for the matter wave theory? This is very strange to me.
>
>  **Response:**
> Thank you for providing valuable suggestions for improving the manuscript. We have cited de Broglie's seminal work on matter wave theory [S.4] at appropriate locations in the manuscript.
>
> [S.4] De Broglie, L., 1924. Recherches sur la théorie des quanta (PhD Thesis). Migration-université en cours d’affectation.
>
> **Question 3:**
> >From the experiments, WDPA for energy prediction works much better than force prediction. Do you have any explanation for this?
>
> **Response:**
> Thank you for providing valuable suggestions for improving the manuscript. The reason why the force prediction performance of the WDPA model does not reach the optimum is that it calculates forces via the negative gradient of energy, which introduces error accumulation. Consequently, its single-step force prediction performance is inferior to models like the Equivariant Transformer (EquiformerV2) that employ direct force fitting. However, the advantage of computing forces through the energy gradient lies in its inherent adherence to physical laws, which enhances the stability of long-time molecular dynamics simulations, as validated in the DPA2 model literature [S.1]. We have elaborated on the relevant content in the original text in Section 4.4, page 8, line 427-468.
>
> [S.1] Zhang, D., Liu, X., Zhang, X., Zhang, C., Cai, C., Bi, H., Du, Y., Qin, X., Peng, A., Huang, J., others, 2024. DPA-2: a large atomic model as a multi-task learner. npj Computational Materials 10, 293. https://doi.org/10.1038/s41524-024-01493-2
>
> **Question 4:**
> >Why do you need Planck's constant here? There is no inherent physical process here, so we can set the scale to whatever we like. In fact, in most DFT calculations we use atomic units where $h = 1$.
>
> **Response:**
> Thank you for providing valuable suggestions for improving the manuscript. Our matter wave network design is principally inspired by matter wave theory rather than strictly adhering to its physical principles. To maintain maximum consistency with the matter wave expression, we retained Planck's constant in our formulation. Furthermore, we conducted supplementary ablation studies that verified the inclusion of Planck's constant enhances the model's predictive performance. For specific details, please refer to our response to Weakness 5.

---

> ### Author Response · Authors · 2025-12-03
> **Response to Reviewer AD9s (Second Round)**
>
> **Weakness 5:**
> >Using sinusoids to encode geometry is not new. atomistic GNNs often use periodic/basis expansions (e.g., Bessel/spherical harmonic bases), and many models use Fourier features or sinusoidal activations. The paper does not benchmark against such periodic encodings, which weakens a novelty claim tied to "periodicity."
>
> **Response:**
> Based on our initial response, to further validate the effectiveness of the matter wave function, ablation studies were conducted on the perovskite oxide dataset FerroEle, and the experimental results were added to the original text in Table A1 in Section A.3.2, page 16, line 810 as follows:
>
> ## Performance Comparison of Predictions
> **(FerroEle Dataset, 1 Million Training Steps)**
>
> | Metrics / Models | DPA2 | WDPA-w | WDPA-p | WDPA-r | **WDPA** |
> | :--- | :---: | :---: | :---: | :---: | :---: |
> | **Energy [meV]** | | | | | |
> | &nbsp;&nbsp;MAE | 22.4 | 20.0 | 19.2 | 19.2 | **17.5** |
> | &nbsp;&nbsp;RMSE | 30.7 | 28.5 | 25.8 | 27.1 | **23.4** |
> | &nbsp;&nbsp;R²(%) | 96.0 | 96.6 | 96.5 | 96.6 | **96.9** |
> | **Energy [meV/atom]** | | | | | |
> | &nbsp;&nbsp;MAE | 0.5 | 0.4 | 0.4 | 0.4 | **0.4** |
> | &nbsp;&nbsp;RMSE | 0.6 | 0.6 | 0.5 | 0.5 | **0.5** |
> | **Force [meV/Å]** | | | | | |
> | &nbsp;&nbsp;MAE | 22.2 | 21.6 | 21.3 | 20.9 | **20.2** |
> | &nbsp;&nbsp;RMSE | 31.4 | 31.0 | 31.6 | 29.6 | **28.7** |
> | &nbsp;&nbsp;R²(%) | 99.8 | 99.8 | 99.8 | 99.8 | **99.8** |
> | **Virial [meV]** | | | | | |
> | &nbsp;&nbsp;MAE | 240.8 | 253.9 | 237.6 | 232.7 | **229.9** |
> | &nbsp;&nbsp;RMSE | 359.8 | 571.1 | 485.0 | 434.8 | **344.9** |
> | &nbsp;&nbsp;R²(%) | 99.6 | 99.5 | 99.5 | 99.6 | **99.7** |
> | **Virial [meV/atom]** | | | | | |
> | &nbsp;&nbsp;MAE | 5.6 | 5.9 | 5.5 | 5.4 | **5.4** |
> | &nbsp;&nbsp;RMSE | 8.4 | 13.8 | 11.5 | 10.3 | **8.1** |
>
> Concurrently, the visualization of the table was updated in Figure A4 in Section A.3.2, page 15, line 774.
>
> **Weakness 9:**
> >There is no molecular dynamics evaluation (e.g., NVE energy drift, long horizon stability, diffusion coefficients). Since the motivation is improved MD, the per frame test error is only a proxy.
>
> **Response:**
> Based on our initial response, molecular dynamics (MD) simulations have been added to the original text. Specifically, using a water molecule (H2O) simulation system, WDPA and DPA2 were employed as the potential energy functions in the LAMMPS molecular dynamics software. The simulations were used to evaluate energy drift, energy fluctuation, and diffusion coefficient, with the latter being compared against both experimental measurements and DFT calculation results (reference [S.1]) in Section 4.5, page 10, line 497-518 as follows:
>
> `4.5 Molecular Dynamics Simulation`
>
> `First, a simulation system containing 64 water molecules (H2O) was constructed, and the WDPA and DPA2 models pretrained on the H2O-PD dataset were employed as the potential energy functions within the LAMMPS molecular dynamics software... The molecular dynamics simulation results are summarized in Table 4.`
>
> ## Molecular Dynamics Simulation Results.
>
> ### Diffusion Coefficient [Å²/ps]
>
> | Parameters         | Ground Truth | PBE   | PBE+vdW | DPA2  | WDPA  |
> | :----------------- | :----------: | :---: | :-----: | :---: | :---: |
> | **330K, 50ps, D**  |    0.186     | 0.015 |  0.041  | 0.078 | 0.081 |
>
> ### Energy Drift (ED) [%]
>
> | Parameters         | DPA2   | WDPA      |
> | :----------------- | :----: | :-------: |
> | **330K, 50ps, D**  | 0.238  | **0.075** |
> | **330K, 5ps, D**   | 0.182  | **0.066** |
> | **275K, 5ps, H**   | -0.204 | **-0.177**|
> | **330K, 5ps, H**   | **-0.077** | 0.116  |
> | **500K, 5ps, H**   | 0.743  | **0.693** |
>
> ### Energy Fluctuation (EF) [%]
>
> | Parameters         | DPA2  | WDPA      |
> | :----------------- | :---: | :-------: |
> | **330K, 50ps, D**  | 0.083 | **0.071** |
> | **330K, 5ps, D**   | 0.049 | **0.047** |
> | **275K, 5ps, H**   | 0.064 | **0.062** |
> | **330K, 5ps, H**   | 0.057 | **0.054** |
> | **500K, 5ps, H**   | 0.207 | **0.206** |
>
> `As shown in Table 4, compared to DPA2, ED and EF of WDPA are consistently closer to zero... Consequently, the diffusion coefficient obtained with WDPA is closer to the experimentally measured values than those derived from PBE, PBE+vdW, or DPA2.`
>
> The content related to the atomic configuration optimization experiments has been moved to the appendix in Section A.4, pages 15-17, lines 806-900.
>
> [S.1] Zheng, L., Chen, M., Sun, Z., Ko, H.-Y., Santra, B., Dhuvad, P., Wu, X., 2018. Structural, electronic, and dynamical properties of liquid water by ab initio molecular dynamics based on SCAN functional within the canonical ensemble. The Journal of Chemical Physics 148.

---

### Official Review · Reviewer_757o · 2025-11-01

**Soundness:** 2
**Presentation:** 2
**Contribution:** 2
**Rating:** 2
**Confidence:** 4

**Summary:**

The paper produces the Matter Wave Deep Potential Atomic (WDPA) model that claims to incorporate physics-based inductive bias into the architecture. This incorporation is given by using the "matter wave feature transformation function", a sinusoidal function as part of the layers. Additionally, the paper provides benchmarks for the resulting model on several datasets, including FerroEle, H2O-PD, and Cathode.

**Strengths:**

Consistent improvements in accuracy over the studied datasets.

A general-purpose module that can be plugged into other architectures.

**Weaknesses:**

Physical justification is nominal. The matter-wave transform reduces to a learnable sinusoid of linear combinations of features and distances. That is indistinguishable in practice from widely used Fourier/Bessel radial–angular bases or positional encodings. The presence of the Planck constant in the matter wave feature transformation function is off the point and seems not to be dimensionally consistent.

In the introduction, the paper claims that the architecture is also built upon Kolmogorov-Arnold Networks, but this connection is missing in the further discussion.

All the presented benchmark datasets are relatively small, with fewer than 10 thousand frames, much smaller compared to such projects as open materials or open molecules with more than 100 million configurations each. Thus, the "large-scale" claim is not justified.

**Questions:**

What is the connection between the proposed model and matter wave theory beyond merely sinusoidal expressions?

What is the connection between the proposed model and Kolmogorov-Arnold Networks?

---

> ### Author Response · Authors · 2025-11-20
>
> **Weakness 1:**
> >Physical justification is nominal. The matter-wave transform reduces to a learnable sinusoid of linear combinations of features and distances. That is indistinguishable in practice from widely used Fourier/Bessel radial–angular bases or positional encodings. The presence of the Planck constant in the matter wave feature transformation function is off the point and seems not to be dimensionally consistent.
>
> **Response:**
> Thank you for providing valuable suggestions for improving the manuscript. Fourier and Bessel radial-angular basis functions, along with positional encoding, serve as effective representation methods, and the matter wave feature transformation function shares certain similarities with them. However, a key distinction lies in the fact that the matter wave feature transformation function explicitly captures the relationship between energy features and spatial features, rather than merely representing spatial features alone. Its design is inspired by the core relational expressions in matter wave theory—namely, the matter wave frequency $f = \frac{E}{h}$ and wavelength $\lambda  = \frac{h}{p}$, where $E$, $p$, and $h$ denote energy, momentum, and Planck’s constant, respectively. By substituting these into a concise sinusoidal wave function, we guide the model to learn underlying physical laws. To preserve the completeness of this formulation, Planck’s constant is retained, as reflected in the matter wave feature transformation function described in the original text. Theoretically, these expressions can be incorporated into any form of wave function, since the core insight lies in the unique relational expressions for matter wave frequency and wavelength, rather than in the specific form of the trial wave function. For instance, one may also adopt non-periodic forms such as wavelet functions, rather than being limited to periodic representations like Fourier basis functions.
>
> To further validate the effectiveness of the matter wave feature transformation function, we replaced it with a Fourier function and decomposed the matter wave expression to separately transform energy and coordinate features or remove Planck's constant, denoted as the WDPA-p model, and added the model description in the original text in Section A.3.2, page 15, line 769-771 as follows:
>
> `4) WDPA-p: Several modifications to the matter wave feature transformation function were explored: replacing it with a Fourier function, decomposing its expression to separately transform energy and coordinate features, and removing Planck's constant.`
>
> We then conducted ablation studies on subsets of the Water Molecules, Perovskite Oxides, and Solid-State Electrolytes datasets—H2O-PD-D, FerroEle-D, and SSE-PBE-D—and supplemented the experimental results in the original text in Table A2 in Section A.3.2, page 16, line 840 as follows:
>
> ## Performance Comparison of Predictions
> **(H2O-PD-D, FerroEle-D, and SSE-PBE-D Datasets, 3,000 Training Steps)**
>
> | Dataset | Energy Type | Metric | DPA2 | WDPA-w | WDPA-p | WDPA-r | WDPA |
> |---------|-------------|--------|------|--------|------------|--------|----------|
> | H2O-PD-D | Energy [meV] | MAE | 218.0 | 251.8 | 211.8 | 196.3 | **186.3** |
> | | Energy [meV] | RMSE | 277.8 | 316.4 | 266.6 | 246.0 | **232.4** |
> | | Energy [meV] | R²(%) | 80.1 | 74.2 | 81.6 | 84.4 | **86.1** |
> | | Energy [meV/atom] | MAE | 1.1 | 1.2 | 1.0 | 1.0 | **0.9** |
> | | Energy [meV/atom] | RMSE | 1.4 | 1.6 | 1.3 | 1.2 | **1.2** |
> | FerroEle-D | Energy [meV] | MAE | 522.1 | 434.5 | 581.0 | 548.8 | **429.1** |
> | | Energy [meV] | RMSE | 686.9 | 542.2 | 710.5 | 688.5 | **536.8** |
> | | Energy [meV] | R²(%) | 88.3 | 92.7 | 85.3 | 88.2 | **92.8** |
> | | Energy [meV/atom] | MAE | 8.7 | 7.2 | 9.6 | 9.1 | **7.1** |
> | | Energy [meV/atom] | RMSE | 11.4 | 8.9 | 11.8 | 11.4 | **8.9** |
> | SSE-PBE-D | Energy [meV] | MAE | 1382.8 | 1594.5 | 1664.2 | 1450.0 | **1295.6** |
> | | Energy [meV] | RMSE | 1654.3 | 1870.0 | 1944.1 | 1729.7 | **1614.9** |
> | | Energy [meV] | R²(%) | 92.3 | 90.1 | 89.2 | 91.3 | **92.6** |
> | | Energy [meV/atom] | MAE | 3.4 | 3.9 | 4.1 | 3.5 | **3.2** |
> | | Energy [meV/atom] | RMSE | 4.1 | 4.6 | 4.8 | 4.2 | **4.0** |
>
> **Weakness 2:**
> >In the introduction, the paper claims that the architecture is also built upon Kolmogorov-Arnold Networks, but this connection is missing in the further discussion.
>
> **Response:**
> Thank you for providing valuable suggestions for improving the manuscript. Our approach is primarily inspired by the Kolmogorov-Arnold Network (KAN) model, where we redesign the learnable activation functions as matter wave feature transformation functions to closely approximate the matter wave feature transformation process in matter wave theory. We have elaborated on the relevant content in the original text in Section 2, page 3, line 144-158.

---

> ### Author Response · Authors · 2025-11-20
>
> **Weakness 3:**
> >All the presented benchmark datasets are relatively small, with fewer than 10 thousand frames, much smaller compared to such projects as open materials or open molecules with more than 100 million configurations each. Thus, the "large-scale" claim is not justified.
>
> **Response:**
> Thank you for providing valuable suggestions for improving the manuscript. The term "Large-Scale" in this work primarily refers to the model's parameter size, its generalizability and extensibility in design, and its foundation based on the Deep Potential Atomic (DPA2) model [S.1], which possesses pretraining-finetuning-distillation capabilities. The current research focuses on integrating prior knowledge from matter wave theory into the neural network design, validated on both fundamental Water Molecules datasets (H2O-PD) and datasets with potential application value, including Perovskite Oxides (FerroEle), Hafnium Dioxide (HfO2), and Oxide Cathodes (Cathode). These datasets are characterized by their relatively large system sizes, with individual atomic configurations ranging from tens to hundreds of atoms. Consequently, DFT labeling costs typically limit these datasets to several tens of thousands of frames. Compared to small-molecule datasets comprising hundreds of thousands to millions of frames—where individual configurations generally contain no more than 20 atoms—the present datasets offer advantages in terms of system complexity and specific application relevance. Future work will involve validation on additional datasets.
>
> [S.1] Zhang, D., Liu, X., Zhang, X., Zhang, C., Cai, C., Bi, H., Du, Y., Qin, X., Peng, A., Huang, J., others, 2024. DPA-2: a large atomic model as a multi-task learner. npj Computational Materials 10, 293. https://doi.org/10.1038/s41524-024-01493-2
>
> **Question 1:**
> >What is the connection between the proposed model and matter wave theory beyond merely sinusoidal expressions?
>
> **Response:**
> Thank you for providing valuable suggestions for improving the manuscript. The matter wave network further approximates the linear superposition principle and self-consistent convergence characteristics of matter wave theory through linear summation and multilayer propagation operations.
>
> **Question 2:**
> >What is the connection between the proposed model and Kolmogorov-Arnold Networks?
>
> **Response:**
> Thank you for providing valuable suggestions for improving the manuscript. The design of the matter wave network is primarily inspired by the Kolmogorov-Arnold Network (KAN) model, where the learnable activation functions are redesigned as matter wave feature transformation functions.

---

> ### Author Response · Authors · 2025-12-03
> **Response to Reviewer 757o (Second Round)**
>
> **Weakness 1:**
> >Physical justification is nominal. The matter-wave transform reduces to a learnable sinusoid of linear combinations of features and distances. That is indistinguishable in practice from widely used Fourier/Bessel radial–angular bases or positional encodings. The presence of the Planck constant in the matter wave feature transformation function is off the point and seems not to be dimensionally consistent.
>
> **Response:**
> Based on our initial response, to further validate the effectiveness of the matter wave function, ablation studies were conducted on the perovskite oxide dataset FerroEle, and the experimental results were added to the original text in Table A1 in Section A.3.2, page 16, line 810 as follows:
>
> ## Performance Comparison of Predictions
> **(FerroEle Dataset, 1 Million Training Steps)**
>
> | Metrics / Models | DPA2 | WDPA-w | WDPA-p | WDPA-r | **WDPA** |
> | :--- | :---: | :---: | :---: | :---: | :---: |
> | **Energy [meV]** | | | | | |
> | &nbsp;&nbsp;MAE | 22.4 | 20.0 | 19.2 | 19.2 | **17.5** |
> | &nbsp;&nbsp;RMSE | 30.7 | 28.5 | 25.8 | 27.1 | **23.4** |
> | &nbsp;&nbsp;R²(%) | 96.0 | 96.6 | 96.5 | 96.6 | **96.9** |
> | **Energy [meV/atom]** | | | | | |
> | &nbsp;&nbsp;MAE | 0.5 | 0.4 | 0.4 | 0.4 | **0.4** |
> | &nbsp;&nbsp;RMSE | 0.6 | 0.6 | 0.5 | 0.5 | **0.5** |
> | **Force [meV/Å]** | | | | | |
> | &nbsp;&nbsp;MAE | 22.2 | 21.6 | 21.3 | 20.9 | **20.2** |
> | &nbsp;&nbsp;RMSE | 31.4 | 31.0 | 31.6 | 29.6 | **28.7** |
> | &nbsp;&nbsp;R²(%) | 99.8 | 99.8 | 99.8 | 99.8 | **99.8** |
> | **Virial [meV]** | | | | | |
> | &nbsp;&nbsp;MAE | 240.8 | 253.9 | 237.6 | 232.7 | **229.9** |
> | &nbsp;&nbsp;RMSE | 359.8 | 571.1 | 485.0 | 434.8 | **344.9** |
> | &nbsp;&nbsp;R²(%) | 99.6 | 99.5 | 99.5 | 99.6 | **99.7** |
> | **Virial [meV/atom]** | | | | | |
> | &nbsp;&nbsp;MAE | 5.6 | 5.9 | 5.5 | 5.4 | **5.4** |
> | &nbsp;&nbsp;RMSE | 8.4 | 13.8 | 11.5 | 10.3 | **8.1** |
>
> Concurrently, the visualization of the table was updated in Figure A4 in Section A.3.2, page 15, line 774.

---

### Meta-Review · Area_Chair_emRf · 2026-01-05

**Summary:**

The paper proposes the Matter Wave Deep Potential Atomic (WDPA) model, which incorporates a matter-wave-inspired feature transformation implemented via sinusoidal functions within the network layers. While all reviewers agree that the paper is clearly written and well presented, they raise several shared concerns, including

1. The proposed feature transformation appears closely related to widely used Fourier feature encodings, and the novelty is not fully convincing.

2. The numerical experiments are conducted on relatively small benchmark datasets, with only modest performance improvements.

3. The paper does not include molecular dynamics (MD) simulation results, which limits the assessment of the model’s practical impact in downstream applications.

**Reviewer Concerns:**

The rebuttal partially addresses the major shared concerns, but not to a sufficient extent. The authors provide an ablation study indicating that using pure Fourier features applied solely to energy or spatial inputs leads to slightly worse performance than the proposed feature design. However, the observed performance gap is small.

Regarding downstream validation, the authors do not provide MD simulation results in the rebuttal. Instead, they present atomic configuration experiments as a simplified proxy, arguing that improved energy prediction accuracy translates to better downstream behavior.

**Reviewer Scores:**

Reviewer 757o would likely retain their original score of 2, as the benchmarks remain relatively small and limited in scope.

Reviewer AD9s would also likely retain their original score of 2, since the rebuttal only partially addresses the core concerns regarding novelty and experimental validation.

Reviewer KQgY would likely maintain their original score of 2, as the main identified weakness is not directly or convincingly addressed in the rebuttal.

---

### Decision · Program_Chairs · 2026-01-26

Reject